

# June 21 and 25, 2015 CMEs interaction's results on Earth's ionosphere and magnetosphere

Somaiyh Sabri[1] and Stefaan Poedts[2,3]

[1]Institute of Geophysics, Faculty of Physics, University of Tehran, Tehran, Iran
[2]Center for mathematical Plasma Astrophysics, Department of Mathematics, KU Leuven, Celestijnenlaan 200B, 3001 Leuven, Belgium
[3]Institute of Physics, University of Maria Curie-Skłodowska, ul. Radziszewskiego 10, PL-20-031 Lublin, Poland

**Correspondence:** Somaiyh Sabri (s.sabri@ut.ac.ir)

**Abstract.** This research utilizes the Gorgon-Space code to simulate the behavior of plasma in the magnetosphere due to its capacity to replicate a vast area. Having a good understanding of the solar wind's characteristics, its relationship with the Earth's magnetic field, the various plasma populations in the Earth's magnetic field, and how they are connected to the ionosphere is greatly dependent on the explanation provided by magnetohydrodynamic (MHD) plasma. Moreover, the European Heliosphere

Forecasting Information Asset (EUHFORIA) serves as a mathematical tool to investigate the movement of coronal mass ejections (CMEs) within the solar wind and, more importantly, determine their estimated arrival time on Earth. To sum up, the research results indicate that the simulation programs EUHFORIA and Gorgon-Space demonstrate a strong correlation in simulating these intricate interactions between the sun and Earth. Additionally, it was observed that when CME1 interacts with Earth, it generates a significant electric potential in the ionosphere. It was discovered that the increased electric potential in the

ionosphere is also responsible for accelerating particles.

## 1 Introduction

Coronal mass ejections (CMEs) happen when a significant amount of plasma is discharged from the sun's corona and released into outer space. Coronal mass ejections (CMEs) are powerful occurrences on the Sun that involve the forceful expulsion of plasma and magnetic fields, resulting in the creation of notable structures. The sudden transformation of magnetic structures,

known as magnetic reconnection, can lead to a substantial release of magnetic energy Sabri et al. (2018); McLaughlin et al. (2018); Sabri et al. (2019, 2020a, b, 2021a, b, 2022, 2023); Kumar et al. (2024). Space weather pertains to the effects of solar activity on Earth as well as on other celestial entities within the solar system. The scientific community acknowledges the growing importance of examining space weather because it affects human activities. In fact, different settings necessitate the application of specific laws of physics. In order to tackle this issue, the ESA Virtual Space Weather Modeling Center

(VSWMC) has enhanced its capabilities to incorporate multiple models for the purpose of predicting the future Poedts et al. (2020).



Solar wind transients like CMEs and corotating interaction regions play a vital role in causing space weather effects, due to their interaction with the coupled magnetosphere-ionosphere system. These can contain large out-of-ecliptic magnetic field components and carry interplanetary shocks at their leading edge. The magneropause experiences strong compression due to a sudden increase in both the density and velocity of solar wind, known as dynamic pressure enhancement (DPE), which results in strong compression of the magneropause. The ground sign of this compression is a sharp, bipolar variation in the horizontal magnetic field Smith et al. (2019). This phenomenon is called the geomagnetic sudden commencement (SC). If it evolves into a geomagnetic storm, it is referred to as storm sudden commencement (SSC) or a sudden impulse (SI) Araki, (1994). SSCs are becoming more acknowledged as a potential danger to power systems in relation to space weather. This is because they have the ability to generate extremely high levels of geomagnetically-induced currents Eastwood et al. (2018).

Furthermore, the magnetosphere's interaction with solar wind also causes a creation of a bow shock up to the magnetopause. The ever-changing nature of the solar wind leads to the constant shifting of the boundaries of the outer magnetosphere as it reacts to variations in pressure and the interplanetary magnetic field (IMF). Different studies suggest that the shape and location of the bow shock primarily depend on the dynamic pressure exerted by the solar wind Merka et al. (2005); Peredo et al. (1995). In empirical models, it is typically presumed that the pressure of the solar wind is consistent throughout the surface of the bow shock Jerab et al. (2005); Merka et al. (2005).

Beyond the usual fluctuations in solar wind, the rapid and substantial movement of boundaries is attributed to distinct formations, especially interplanetary shocks Dryer et al. (1967); Grib et al. (1979). Many studies have recreated and studied the impact of observed interplanetary shocks on the Earth's magnetosphere. These studies have discovered that the interaction between the shock and the bow shock leads to the formation of a sequence of discontinuities Pallocchia (2013); Prech et al. (2008). These have been proven to generate further shocks as they travel through the shock and sheath, which in turn cause deviations and reflections off the magnetopause. Simulations by Samsonov et al. (2007) reveal that there are inward movements of the Earth's bow shock and magnetopause when the interplanetary shock hits. However, there are also outward movements caused by a reflected shock. It was intriguing to note that spacecraft have witnessed comparable findings Pallocchia (2013).

In general, empirical models have limitations in capturing the motion of shocks that occur in response to rapid changes in the solar wind upstream. Specifically, there is limited comprehension regarding the point at which these statistical models are no longer suitable, and it is not evident how these models deviate from reality when considering the dynamic transition of boundaries towards new equilibrium states. A convenient method for analyzing the entire system accurately is by utilizing global magnetohydrodynamic simulations, which can simulate the system as a whole in a consistent manner. In this study, the Gorgon code is used to conduct global MHD simulations of Earth's magnetosphere while considering the changing solar wind conditions Chittenden et al. (2004); Ciardi et al. (2007); Mejnertsen et al. (2016).

The Earth's magnetosphere poses a significant difficulty for researchers because it is not uniform in structure and constantly changes. Moreover, it is crucial to have precise understanding of the solar wind's actions and how it interacts with the magnetospheric field and various plasma groups within the magnetosphere. Additionally, the connection between these





plasma populations and the ionosphere is essential when modeling plasma dynamics in the magnetosphere. We need to use the Gorgon-Space code as it enables us to describe the magnetohydrodynamic (MHD) plasma in the extensive simulation area.

The primary connection between the ionosphere and magnetosphere is through the field-aligned current (FAC), which is linked to the vorticity in the MHD slab. The ionosphere is the upper part of the Earth's atmosphere, located between 60 and 1000 km in altitude. The conditions in the magnetosphere are crucial for the physical processes occurring in the ionosphere. So, we are also contemplating the exploration of the magnetosphere-ionosphere system. Physics-based models can assess the state of the magnetosphere-ionosphere by considering the solar wind and interplanetary magnetic field. In this sentence, we examine

temporary changes in the field-aligned currents (FAC) in the ionosphere.

Since we studied two defined CMEs in our previous paper and investigated their propagation at the space and find that they have interaction with Earth. Then, the objective of our study is to investigate the interaction of two coronal mass ejections (CMEs) with the Earth and its following results on the ionosphere and magnetosphere of the Earth. In this research, we utilize

Gorgon-Space to explore how Earth's magnetosphere reacts to the interaction with CMEs and study the impact of CMEs on the magnetosphere. In conclusion, the study examines how the ionosphere reacts to the interaction between Coronal Mass Ejections (CMEs) and the magnetosphere. It also evaluates the consequential effects on the ionosphere, including induced current density, which has a significant impact on communication and space weather.

## 2  Method

### 2.1  The Gorgon MHD code

Gorgon is a 3D magnetohydrodynamic code, initially developed for studying high energy, collisional plasma interactions such as Z-pinches Chittenden et al. (2004); Jennings et al. (2010), laser-plasma interactions Smith et al. (2007), and magnetic tower jets Ciardi et al. (2007). It has recently been adapted to simulate planetary magnetospheres and their interaction with

the solar wind Mejnertsen et al. (2016). Gorgon employs a comprehensive and clear representation of the resistive semiconservative Magneto-Hydrodynamic (MHD) equations for a plasma that is fully ionized. The majority of its primary features are specifically developed for intricate collisional and resistive Magnetohydrodynamics (MHD) occurrences Ciardi et al. (2007). Gorgon-Space has been created specifically to tackle the challenge of conducting global magnetospheric simulations for space weather and space physics. Therefore, it currently represents a distinct area of enhancement and has been optimized and eval-

uated for particular use cases. However, unlike other global magnetospheric codes, Gorgon-Space calculates the magnetic vector potential instead of the magnetic field. This approach guarantees that the field remains divergence-free Mejnertsen et al. (2018); Eggington et al. (2020). Gorgon-Space utilizes a definite-sized 3D Cartesian grid to solve the MHD equations. Moreover, Gorgon possesses a highly advanced capability to scale linearly and parallelly to numerous cores, based on fully explicit numerical methods. It well found the impact of solar wind on various important parameters in the tail, such as magnetic field,

field aligned current, plasma velocity, and plasma pressure. These global quantities are reliant on the equilibrium of the mag-



netosphere at a large scale, which is believed to be accurately explained by the magnetohydrodynamics (MHD) approach.

Gorgon uses a fully explicit Eulerian formulation of the resistive semiconservative MHD equations for a fully ionized plasma as

$$\frac{\partial \rho}{\partial t} + \nabla \cdot (\rho \mathbf{V}) = 0, \tag{1}$$

$$\rho \left[ \frac{\partial \mathbf{V}}{\partial t} + (\mathbf{V} \cdot \nabla)\mathbf{V} \right] = \left( \frac{1}{\mu} \nabla \times \mathbf{B} \right) \times \mathbf{B} - \nabla(P_e + P_p), \tag{2}$$

$$\frac{\partial \mathbf{B}}{\partial t} = \nabla \times (\mathbf{V} \times \mathbf{B}) + \eta \nabla^2 \mathbf{B}, \tag{3}$$

$$\frac{\partial \epsilon_p}{\partial t} + \nabla \cdot (\epsilon_p \mathbf{V}) = -P_p \nabla \cdot (\mathbf{V}) - \Delta_{pe}, \tag{4}$$

$$\frac{\partial \epsilon_e}{\partial t} + \nabla \cdot (\epsilon_e \mathbf{V}) = -P_e \nabla \cdot (\mathbf{V}) + \Delta_{pe} - \Lambda + \eta |\mathbf{J}^2|, \tag{5}$$

The plasma density, plasma velocity, magnetic field, proton internal energy density, electron internal energy density and thermal plasma pressure are represented by the variables $\rho$, $\mathbf{V}$, $\mathbf{B}$ $\epsilon_p$, $\epsilon_e$, and $P$ respectively. The values $\mu$, $\eta$, and $\gamma = 5/3$ represent the magnetic permeability, the magnetic diffusivity, and the ratio of specific heats at constant pressure and volume, respectively. The permeability of vacuum is considered i.e. $\mu = 4\pi \times 10^{-7}$ Hm$^{-1}$. The deBar correction, known as Von Neumann artificial viscosity, is utilized to properly detect and enhance the conservation of energy during the presence of shocks Benson (1992). Gorgon makes use of second-order Van Leer advection to solve the advection terms in equations. It also employs a variable time step, which automatically satisfies the relevant Courant conditions.

Due to its roots in high-energy plasma physics, the MHD formulation in Gorgon is atypical. It treats the electron and proton internal energy equations separately, allowing them to be out of thermodynamic equilibrium. These equations include terms for ohmic heating $\eta |\mathbf{J}^2|$, optically thin radiation losses $\Lambda$, and electron-proton energy exchange $\Delta_{pe}$. It must be noted that the choice of solving for the internal energy, rather than the total energy, means that the total energy is not strictly conserved. This mitigates effects such as negative pressures occurring in regions of high magnetic field strength.

## 3 Numerical results and discussion

The $K-$index quantifies disturbances in the Earth's magnetic field in the horizontal orientation and is classified with a numerical score ranging from 0 to 9. During peaceful situations, the Kp value is around 1. However, during severe geomagnetic





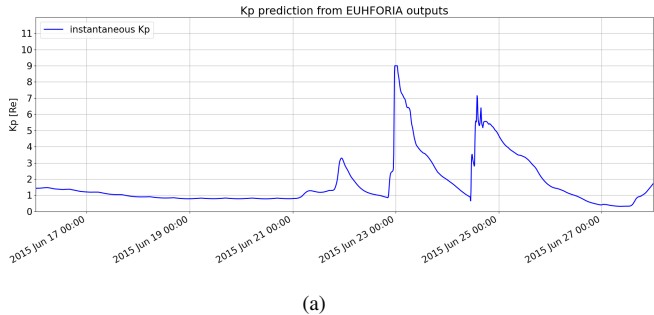

(a)

**Figure 1.** Kp index calculated by the EUHFORIA.

storms, it is about 9. The Kp index is a worldwide evaluation of the K index, which is determined by studying information collected from ground-based magnetometers Bartels & Veldkamp (1949). When it comes to the actual situation, Kp acts as an exceptional indicator for any anomalies in the Earth's magnetic field. Currently, Kp is important as it measures the amount of energy transferred from the solar wind to Earth. It is used by space weather services almost immediately. In our previous study,

? utilized EUHFORIA to examine how two chosen CMEs spread and interacted with Earth.

The fluctuations in the Kp index predicted by the EUHFORIA numerical model matched the actual Kp index observations made by GFZ. This statement affirms that EUHFORIA possesses the ability to precisely ascertain and potentially predict the influence of CMEs on our planet. This representation also highlighted the dominant influence of CME1 on Earth. Moreover,

because of the existence of multiple coronal mass ejections (CMEs) near CME1, it is likely that this expected severe storm is linked to the interaction of several interplanetary coronal mass ejections (ICMEs). However, it was deduced that CME2 weakened and arrived on Earth at a later moment, consequently not causing significant effects on the planet. In this research, we have been investigating how two specific CMEs impact the Earth's magnetosphere and ionosphere. This has been done using the Gorgon Space code to show which CMEs illustrates the main effects on the Earth and whether the prediction with

EUHFORIA code based on the amount of the kp index is valuable or not. The simulation provide a wealth of information about the local plasma parameters controlling pressure and particle acceleration in the Earth's magnetosphere.

### 3.1 Gorgon-Space

A systematic study was carried out to demonstrate the worldwide fluctuations of the magnetic field through magnetic obser-

vations. These differences were credited to the influence of solar wind discontinuities on both the Earth's magnetosphere and ionosphere. Magnetic reconnection is the main process through which plasma and energy from the solar wind are being introduced into the magnetosphere. However, there are also cases where viscous-like effects can play a role, especially when the magnetic setup is unsuitable for reconnection. In this part, we have concentrated on examining how the arrival of CMEs affects





the Earth. The Gorgon-Space code is utilized in this specific situation.


The configuration of the simulation domain can be observed in Fig 2. When the solar wind interacts with the Earth's magnetosphere, it generates a bow shock wave that forms ahead of the magnetopause. This results in the creation of a turbulent region known as the magnetosheath situated between them Lucek et al. (2005); Burgess & Scholer (2013). In order to enable the plasma to move away from the planet, such as when the solar wind is redirected around the magnetopause or when magnetospheric plasma is expelled towards the tail, it is necessary to implement free-flow at the outer boundaries of the simulation domain. One exception to this rule occurs at the side of the box facing the sun, specifically on the left side of Earth. This is where the simulation receives and relies on the solar wind inflow to serve as the main source of plasma.

To allow the plasma to leave the Earth's vicinity, it is necessary to incorporate free-flow outside the simulation area. In simpler terms, when the flow of charged particles from the sun is directed away from the boundary of a magnetic field, the plasma in the magnetic field is expelled towards the center. It should be acknowledged that on the left side of the Earth, there is a box with its sunny side facing towards the Sun. This box serves as the main entry point for the solar wind, which is the primary provider of plasma in the simulation. The solar wind's force, speed, and magnetic field can be established for the incoming plasma and analyzed for its interaction with the magnetic field of the planet.


### 3.1.1 Magnetospheric Response

The magnetosphere is influenced by interactions with the solar wind, which can be either viscous or pressure-related according to Newell et al. (2008). Fig. 3 represents the initial shape of the magnetosphere at the start of the simulation, which correlates with the arrival time of the first Coronal Mass Ejection (CME) at Earth on June 23, 2015. This depiction is valid for the entire analysis period of 25 hours. The text demonstrates three elements of the magnetic field as well as their corresponding values in nanoteslas (nT). It also portrays the velocity components and their values in kilometers per second (km/s), the number density in units of per cubic centimeter ($1/cm^3$), and the temperature in electron volts (eV). It shows the specified limits in divisions along the path of the Earth around the Sun and the planes of noon and midnight. The magnetosphere's features have been altered completely by the simulation.


As indicated in the velocity panel of Fig. 3, the y and z components of the solar wind velocity are close to zero. This means that the solar wind reaching Earth only has the x component of velocity, which is directed towards the Sun-Earth line. Decrease in the density of particles and the rise in temperature, which can be seen in Fig. 3, may be connected to the formation of the bow shock. It should be acknowledged that ICMEs lead to the fastest speed and the least negative $B_z$ component at the Earth's orbit. As a result, ICMEs are considered to be the underlying cause of all significant geomagnetic storms that have a $K_p$ index greater than 7. According to Fig. 1, there is a peak in the $K_p$ values between the 22nd and 24th of June, with a value of over 7. Based on this, it is anticipated that the initial coronal mass ejection (CME) will reach. On 23 June, there is a possibility that





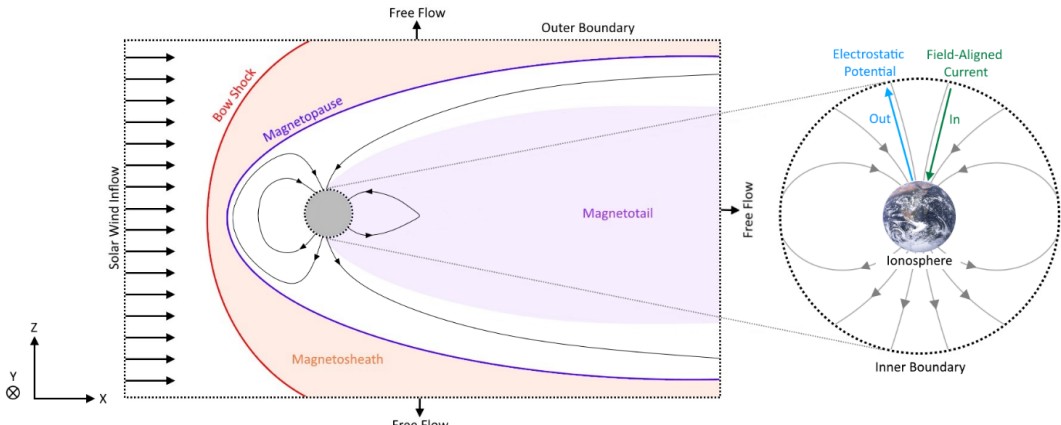

**Figure 2.** The simulation domain setup in Gorgon-Space.

Earth may be hit by an ICME (interplanetary coronal mass ejection), which in turn could lead to a significant storm. Additionally, due to the presence of numerous CMEs between the dates of June 19 and June 23, 2015, it is possible that this predicted
storm is connected to the interaction of multiple ICMEs.

Fig. 4 shows the evolution of thermal pressure during the arriving of CME1 at the magnetosphere, represented by black lines as magnetic field lines. Fig. 4 illustrates diagrams of the magnetosphere, showcasing the pressure and arrangement of open magnetic field lines on both the X-Z plane and the X-Y plane. Afterward, it enables us to display the three-dimensional
structure of the magnetosphere's dynamo regions. The magnetopause's location and the areas where reconnection occurs on the day and night sides are shown. The magnetopause can be defined as the point where the lowest level of solar wind enters the magnetosphere. Panel (a) of Fig. 4 displays the Earth's magnetosphere pressure at the time of the initial occurrence of a coronal mass ejection (CME) from the Sun. Diagram (b) in Fig. 4 depicts the magnetosphere's pressure upon the arrival of the initial coronal mass ejection (CME) at Earth. Comparing panels (a) and (b) reveals that the interaction between solar wind and
the Earth leads to increased pressure and mainly alters the configuration of the magnetic field, particularly in the $XY$ plane. This occurrence may be connected to the magnetic reconnection phenomenon which occurs when the solar wind interacts with Earth's magnetic field. This leads to variations in the magnetic field structure, which is free from any significant obstruction along the Sun-Earth direction, particularly upon the initial arrival of the CME at Earth. It is imperative to accurately predict the expected arrival time of CMEs at Earth, that omits the shielding effect of the magnetosphere.


Fig. 5 illustrates the profiles of pressure and magnetic field lines as the CME2 reaches Earth. By examining Figs. 4 and 5, it can be observed that CME1 causes notable changes in pressure and alters the structure of magnetic field lines. On the other hand, in Fig. 5, it is evident that the magnetosphere retains its primary function as a shield and maintains its key properties. In simpler terms, the magnetopause and bow shock of the CME1 experience significant changes in their size and shape, just as





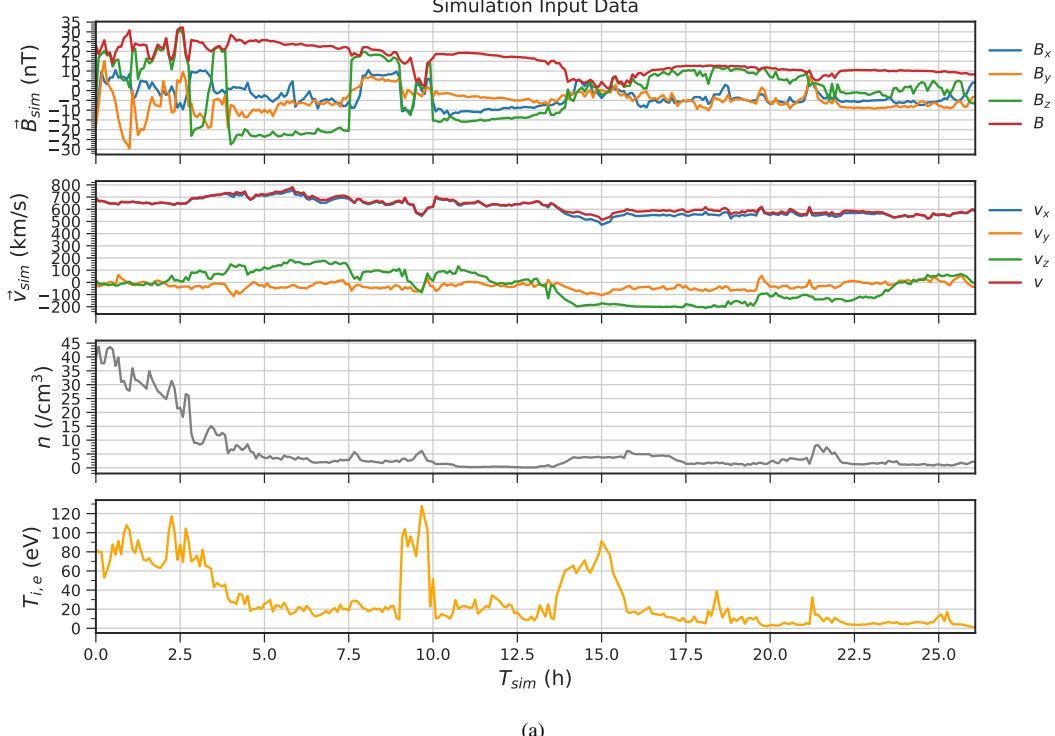

(a)

**Figure 3.** Input Data for Gorgon-Space

predicted.

The primary function of the bow shock is to slow down and deflect the high-speed solar wind as it encounters the magnetosphere. The creation of the foreshock occurs when particles are reflected at the shock, resulting in various interactions between waves and particles and the acceleration of particles Eastwood et al. (2005). We have been making efforts to study

the acceleration of particles through the interaction of the solar wind with the magnetosphere.

The figure displayed in Fig. 6 depicts the changes in solar wind velocity caused by the CME1 at the magnetosphere. The first panel of Fig. 6 shows that the wind speed at the time of CME1 happening on the Sun is not noteworthy. Additionally, the magnetic field formation of the magnetosphere has a consistent shape and can fulfill its main role as a protective barrier. Panels

(b) to (d) depict the wind speed and magnetic configuration of the magnetosphere upon the arrival of CME1 at Earth. Upon initial observation, it is apparent that these three panels depict a notable degree of speed and an irregular form of the magnetic structure resulting from the interaction between the solar wind and the magnetosphere. One can infer that the energy from the solar wind is transferred to the Earth's magnetosphere through a process called magnetic reconnection. This occurs when the



interplanetary magnetic field connects with the Earth's magnetic field at the magnetopause on the dayside of the planet.


Fig. 7 depicts the solar wind velocity and magnetic field configuration upon the arrival of CME2 at Earth. It is evident that CME2 does not lead to the high plasma velocity at the magnetosphere. Besides, in Fig. 7, in X-Z plane the closed field regions get larger on nightside that could be due to a reduction in open flux content in the magnetosphere. This might lead to a decrease in geomagnetic activity and smaller polar caps in the ionosphere.


The rate of the magnetic reconnection and also how much energy is transferred is influenced by the speed at which the solar wind flows, the strength of the magnetic field, and its orientation. Subsequently, it was discovered that CME1 triggers a significant tempest on Earth and is accompanied by changes in the magnetic arrangement, increased speed and pressure of the solar wind enveloping the Earth, as previously anticipated in Fig. 1.


Figs. 5 showcases the changes in pressure and magnetic field in the vicinity of the Earth upon the arrival of CME2. As it was also shown in Fig. 1, there was no indication of a significant magnetic storm at the time when CME2 arrived. It can shed light on impact of the background solar wind that is the same in both CMEs, then it is not the primary factor in causing magnetic storms. Hence, it can be deduced that the nature of the CME and its interaction with other CMEs are crucial factors in determining the geomagnetic impact of CMEs interacting with the Earth.


$$j = \frac{(j.B)B}{|B|^2}, \tag{6}$$

### 3.1.2 Ionospheric Response

In order to investigate the connection between the magnetosphere and ionosphere, an individual module focused on the iono-
sphere is employed. Field-aligned currents are computed at the initial boundary of the simulation and then transferred onto a distinct spherical grid located on the ionosphere, following the dipole field lines. A thin-shell approximation is used to solve the thin-shell Ohm's law and derive an electric potential. Each ionospheric grid cell has a designated conductance, which can be calculated using empirical relationships that consider solar EUV ionization and the underlying auroral conductance. Alternatively, the conductance can also be set as a constant value. Our approach involves finding the solution for the potential
and then using it to create a map that corresponds to the inner boundary. This map is reintroduced as a boundary condition in the MHD simulation. The module for the ionosphere is responsible for carrying out the required mapping, interpolation, and calculations to provide the inner boundary condition for the flow. The ionospheric model utilizes spherical coordinates to allow for the inclusion of larger spatial scales.

The ionospheric effects of the better interaction between the solar wind and reconnection on the dayside are the increase of open flux in the polar cap and the creation of convective flows. This may be connected to the expansion of the boundary




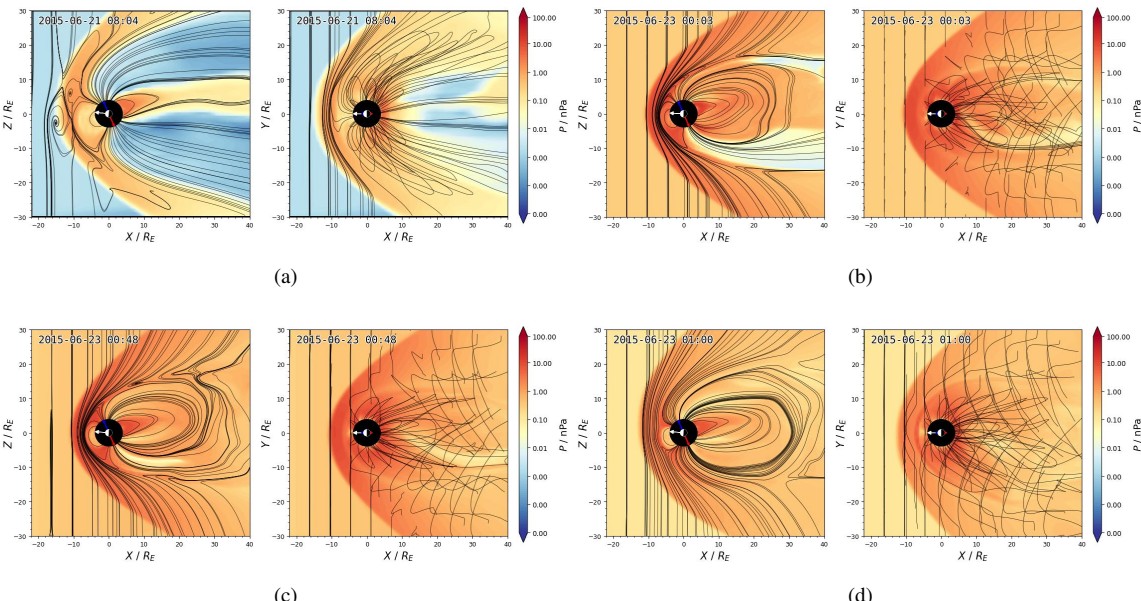

**Figure 4.** Snapshot of solar wind pressure from the MHD simulation with Gorgon-Space. It depicts the arrival time of the first CME that started at 21 June, 2015 at 08:02 and arrived at the Earth on 23 June, 2015 at 00:03 UT in the noon-midnight plane.

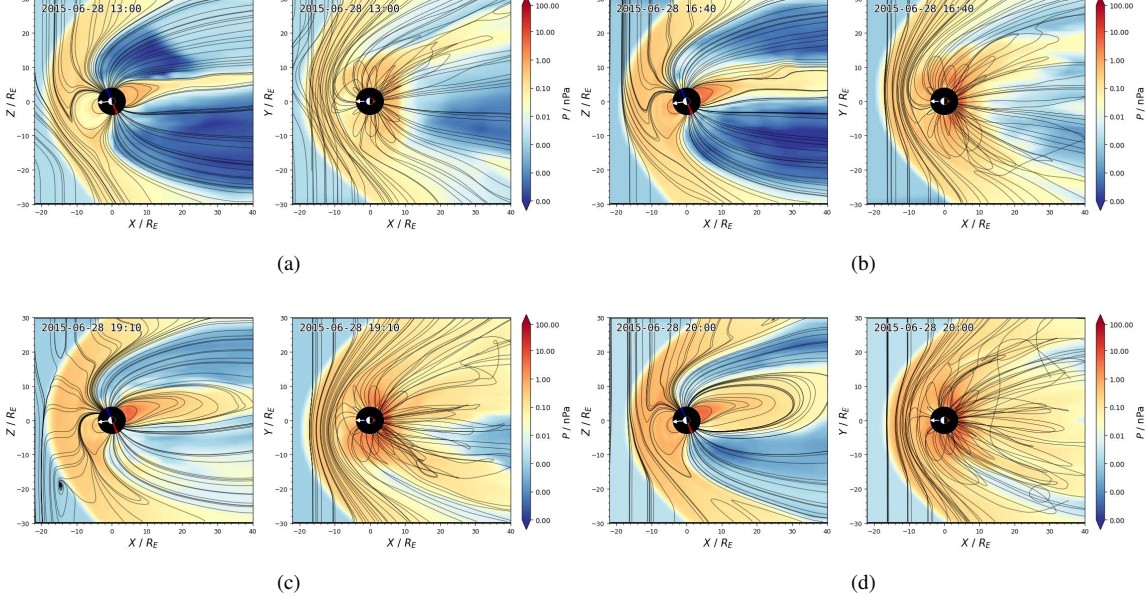

**Figure 5.** Snapshot of solar wind pressure from the MHD simulation with Gorgon-Space. It depicts the arrival time of the second CME that started at 25 June, 2015 at 14:53 and arrived at the Earth on 28 June, 2015 at 12:52 UT in the noon-midnight plane.





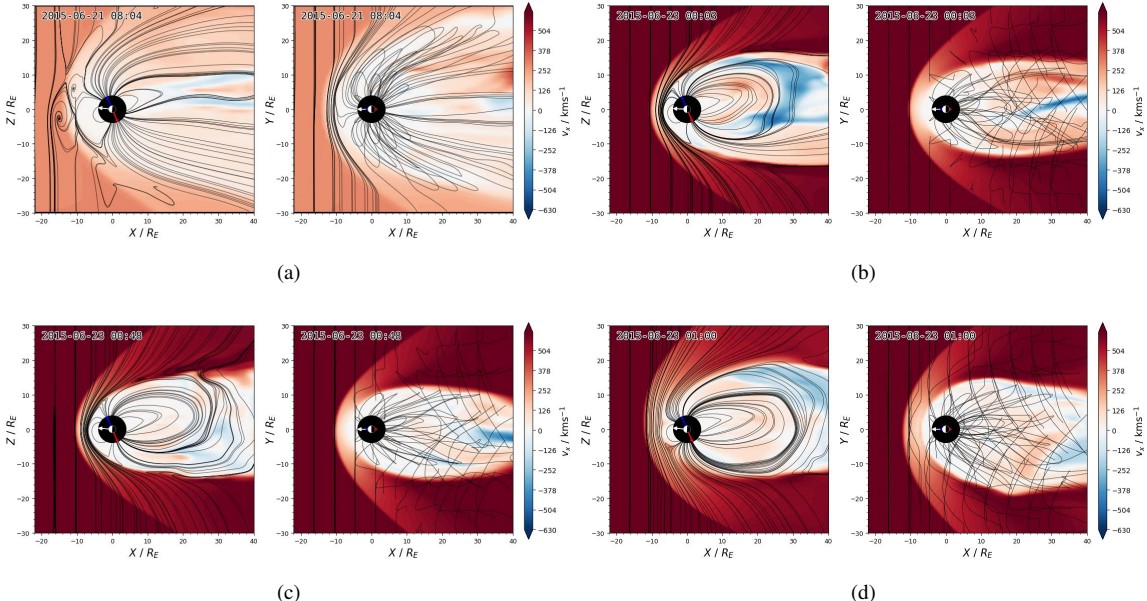

**Figure 6.** Snapshot of solar wind velocity from the MHD simulation with Gorgon-Space. It depicts the arrival time of the first CME that started at 21 June, 2015 at 08:02 and arrived at the Earth on 23 June, 2015 at 00:03 UT in the noon-midnight plane.

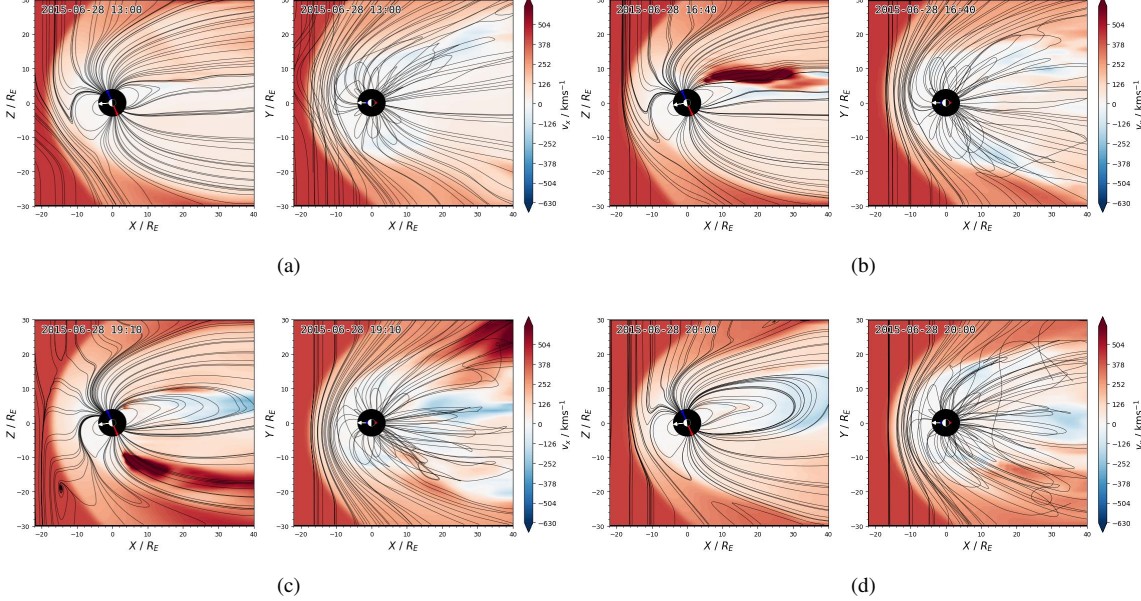

**Figure 7.** Snapshot of solar wind velocity from the MHD simulation with Gorgon-Space. It depicts the arrival time of the second CME that started at 25 June, 2015 at 14:52 and arrived at the Earth on 28 June, 2015 at 12:52 UT in the noon-midnight plane.





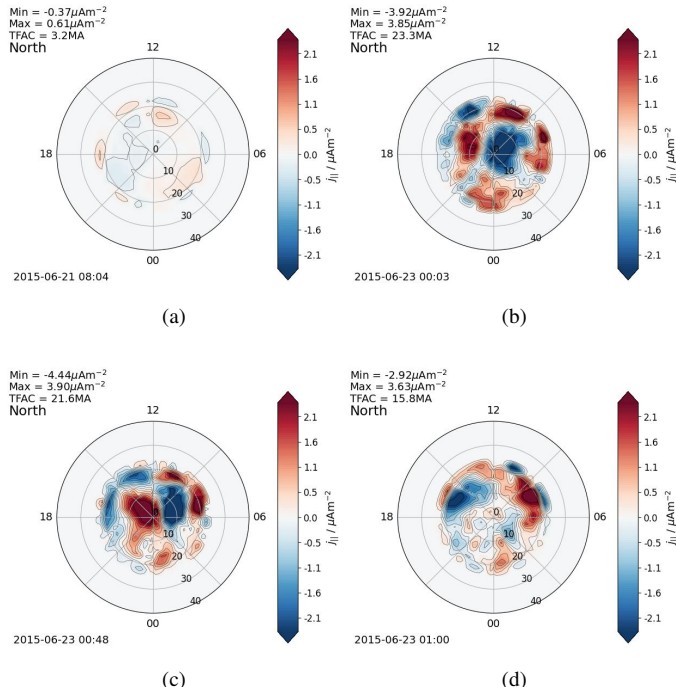

**Figure 8.** Snapshot of the ionospheric field-aligned current density in the Northern Hemisphere.

between open and closed field lines on the dayside. In reality, whenever the magnetic field is distorted or rotated in a way that it is not curl-free in terms of mathematics, it results in a flow of electric current. Equation 7 provides the definition for the current density that is aligned with the magnetic field direction. The current densities that align with the magnetic field lines in the ionosphere, which has resistance, are pulled by the Lorentz force ($J \times B$). This force acts on the plasma and is the main driver of the drag experienced by neutrals on the ionosphere. The magnetosphere contains a force referred to as $J \times B$ which contributes to the flow of current density through the ionosphere. As a result of the stresses caused by the field-aligned currents, which extend from the outer magnetosphere to the ionosphere, a complete flux tube can be transported around the magnetosphere.

The accumulation of current density aligned with the magnetic field can be attributed to the interaction between the solar wind and the magnetosphere. The predominant energy transfer and alignment of currents in the ionosphere occurred primarily due to the boundary conditions. In order to predict space weather accurately, it is necessary to have a solid grasp of the physics behind these coupling processes. Determining the accuracy of MHD codes in replicating the electric fields in the plasma sheet when the solar winds interact with the magnetosphere is of utmost significance in Space Weather applications.



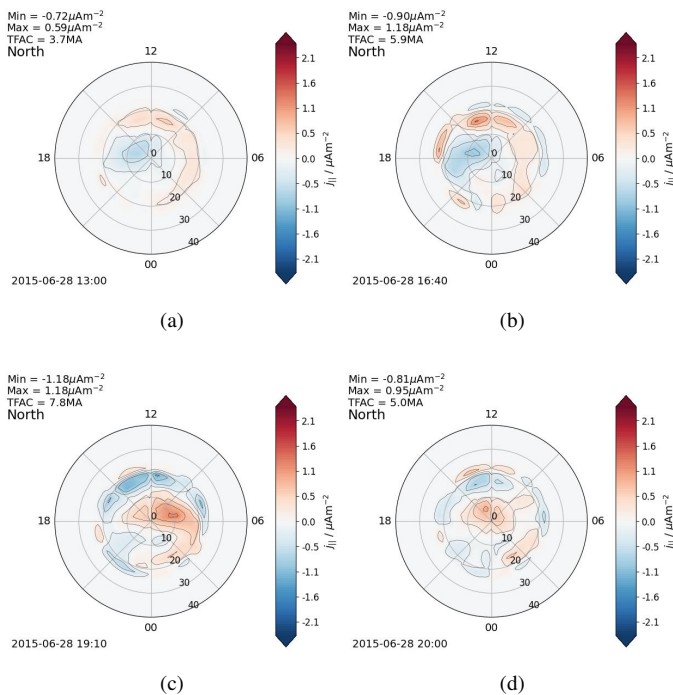

**Figure 9.** Snapshot of the ionospheric field-aligned current density in the Northern Hemisphere due to the interaction of the CME2.

Fig. 8 presents a visualization of the build-up of current density aligned with the magnetic field in the Northern ionosphere due to the arrival of CME1 at Earth. In panel (a) of the Fig. 8, the ionospheric field-aligned current is illustrated during the initiation of the first coronal mass ejection (CME) on June 21, 2015 at 08:02. The colors blue and red show currents aligned

with the field that are directed upward and downward, respectively. It is evident that during the launch of CME on the Sun, there is no noteworthy flow of electric current at the ionosphere. Panel (b) of Fig. 8 demonstrates the field-aligned current in the ionosphere during the initial arrival of the CME at the Earth on June 23, 2015, at 00:03 UT. By examining two panels (a) and (b) in Fig. 8, it becomes evident that when CME1 reaches the Earth, there is a substantial build-up of current density in the ionosphere. This accumulation leads to a total field-aligned current of approximately 23 million amperes. The outcome

reveals a concentrated region of current density buildup in both hemispheres. Actually, the energy for FACs originates from the magnetosphere and can be determined by computing the value of $E.J$. If there is a negative quantity, the energy of the plasma is transformed into electromagnetic energy, which then enhances the FACs. Panels (c) and (d) of Fig. 8 show a decrease in the magnitude of current density and its movement from the zero latitude of the ionosphere towards a latitude ranging from 10 to 20 degrees. In general, CME1 causes magnetic storms when it interacts with Earth's magnetosphere, and this can result

in electrical currents in the Earth's crust that have the potential to cause damage to installations (Pulkkinen, , 2007; Pirjola, , 2005). Furthermore, the fourth panel in Fig. 8 reveals the greatest imbalance in the accumulation of current density between the Northern and Southern hemispheres. This probably originates from the electric field generated by convection itself. To




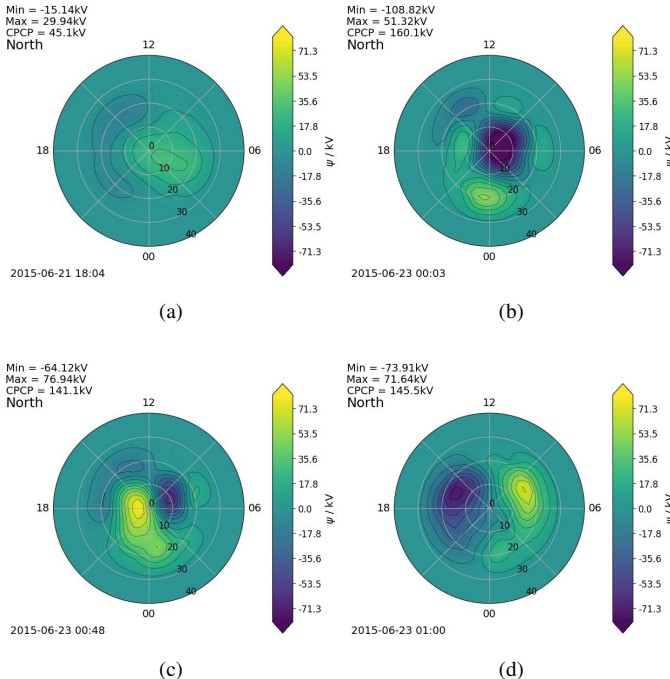

**Figure 10.** Ionospheric response electric potential during the happening of the first CME and its arriving at the Earth. Yellow indicates the positive flux, and blue means negative flux.

provide further explanation, the distance that convection travels along the northern open field lines is greater in comparison to the southern field lines.


Fig. 9 illustrates the build-up of current density in the ionosphere upon the arrival of CME2 at Earth. It is evident that this CME does not lead to the significant current density accumulations at the ionosphere that was expected in Fig. 1. Although the speed of CME2 was greater than that of CME1, it can be inferred that the launching velocity of CMEs does not determine their subsequent damage or outcomes. Since the occurrence of CME1 coincides with other CMEs, it is possible to infer that
the interaction between CMEs is the primary factor influencing their geomagnetic effects.

The Fig. 10 shows the electric potential response in the ionosphere. The illustration shows the cross-polar cap electric potential (CPCP) in the ionosphere, which is widely used to assess the worldwide convection intensity. This displays the disparities between the highest and lowest points of electric potential in the ionosphere. CPCP was initially derived from the integral rate
of dayside-merging and can be employed as an estimate for the key parameter of dayside-merging, according to Gordeev et al. (2015).





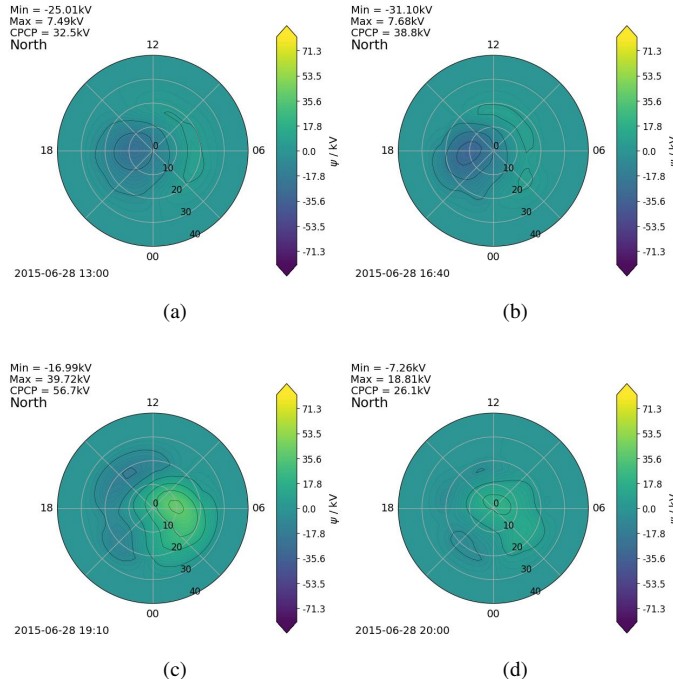

**Figure 11.** Ionospheric response electric potential during the arriving of the CME2 at the Earth. Yellow indicates the positive flux, and blue means negative flux.

The first panel in Fig. 10 shows the level of the electric potential in the ionosphere at the onset of the initial coronal mass ejection (CME1). Panel (b) of Fig. 10 displays the ionospheric electric potential reaction during the initial arrival of the first coronal mass ejection (CME1) at the Earth. Comparing both panels helps to determine the impact of solar wind on Earth, specifically resulting in an electric potential value of approximately $160kV$ in the ionosphere. The elevated electric potential in the ionosphere, which may result from the substorms triggered by the interaction between the solar wind and Earth, can also lead to particle acceleration. This phenomena was pursued in previous section by illustrating Figs. 6, 7, and shown that CME2 does not lead to the high plasma velocity at the magnetosphere. The figure in Fig. 11 illustrates the electric potential in the ionosphere when CME2 reaches the Earth. As predicted, the arrival of CME2 at Earth does not cause any notable electric potential in the ionosphere.

## 4 Conclusions

Because of their significance in both science and society, studies on CMEs become highly important. CMEs are responsible for causing significant geomagnetic storms and generating solar energetic particles (SEPs). Geomagnetic storms have the potential to generate particles in radiation belts, which can impact satellites, while solar energetic particles (SEPs) have the capability



to harm spacecraft. Consequently, it is crucial to examine the characteristics of CMEs, such as their movement and interaction while being influenced by the surrounding solar wind.

As was explained in Sabri et al. (2024), the selected CMEs occurred on June 21 and June 25, 2015. These two CMEs had distinct structures, one being a full halo CME and the other being a limb CME. It is important to mention that there were other coronal mass ejections (CMEs) occurring around CME1. These interactions between the CMEs could have had a notable impact on Earth's magnetosphere and ionosphere that was investigated in this study.

Global MHD models are important tools in the field of space weather, serving as powerful researches and predictive tools that were based on physics. CMEs and solar winds play a crucial role in causing changes in the magnetosphere and ionosphere, making them an essential aspect of space weather. We utilized a numerical MHD model with three-dimensional time-dependence to study expansive solar wind interaction with Earth and its following effects on the Earth's magnetosphere and ionosphere.

The Gorgon-Space code was used to determine the impact of the CMEs on the Earth's magnetosphere and ionosphere. This was done by running the code at the exact times when both CMEs reached the Earth that was found by EUHFORIA Sabri et al. (2024). Gorgon-Space is a code that models the interaction between the magnetosphere and ionosphere on a global scale. It lets us to know how the solar wind resulting from the CMEs interacts with the magnetosphere, and how it alters Earth's magnetic structure. Furthermore, the resulting effects of the interactions of the interplanetary magnetic field and Earth's magnetic field

on the ionosphere and resulting field-aligned current density accumulations were pursued.

     In conclusion, we have achieved the following key findings.:

1. The Earth's magnetosphere experienced increased pressure and the magnetic field's structure, particularly at the $XY$
plane, was mainly altered due to the interaction with CME1. It could be related on the occurrence of magnetic reconnection as a result of the interaction between the Earth's magnetic field and solar wind. In the case of CME1, the size and shape of the magnetopause and bow shock changed considerably.

2. The arrival of CME1 on Earth led to a significant buildup of current density in the ionosphere, with a total field-aligned current of nearly 23 million amperes. This resulted in a highly concentrated area of electric current density on both
hemispheres. CME1 caused magnetic storms when it interacted with the Earth's magnetosphere, leading to a buildup of high current density in the ionosphere.

3. It was found a greatest imbalance in the accumulation of current density between the Northern and Southern hemispheres, due to the interaction of CME1 with the Earth. This probably originates from the electric field generated by convection itself.





4. It was illustrated that launching velocity of CMEs does not determine their subsequent damage or outcomes. Since the occurrence of CME1 coincides with other CMEs, it was inferred that the interaction between CMEs was the primary factor influencing their geomagnetic effects.

5. CME1 interaction with Earth specifically results in an electric potential value of approximately $160kV$ in the ionosphere. It was found that the elevated electric potential in the ionosphere also leads to the particle acceleration.

6. Agreement of the CME1 and CME2 results in interplanetary space (by EUHFORIA), magnetosphere and also ionosphere (by EUHFORIA and Gorgon-space) depicted that two mentioned simulation codes are in good agreement.

*Code and data availability.*    We used EUHFORIA code to numerical study that is publicly available vie the VSWMC in https://euhforia.com/. Besides, we also used Gorgon Spave code to stimulate the effect of the CMEs on Earth.

*Author contributions.*    Authors have contribution to write the manuscript.

*Competing interests.*    The contributer author has declared that none of the authors has any competing interest.

*Acknowledgements.*    For the computations we used the infrastructure of the VSC−Flemish Supercomputer Center, funded by the Hercules foundation and the Flemish Government−department EWI.



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
