# Peer review of "June 21 and 25, 2015 CMEs interaction's results on Earth's ionosphere and magnetosphere"

_EGUsphere, 2024_

## Referee Comment (RC2)

Reviewer report of "June 21 and 25, 2015 CMEs interaction's results on Earth's ionosphere and magnetosphere" by Somaiyh Sabri and Stefaan Poedts

This paper discusses magnetosphere and ionosphere disturbances associated with abrupt changes in solar wind parameters, such as CMEs, using the Gorgon Space model, a specialized MHD code for high-energy plasmas.

The referee judges that this manuscript is completely incomplete as a research paper, given the points listed below. Therefore, the referee recommends that this paper cannot be accepted for publication in the present form.

**Summary of the review**

- 1. The authors appear to lack knowledge of space science. Before writing their paper, they should fully understand the characteristics of the disturbances caused by CMEs in the magnetosphere-ionosphere system and summarize this in the introduction. They should then point out the shortcomings of previous research and explain the novelty and importance of their research.
- 2. Research papers should be able to be retested by third parties, and to this end, the input data used in the simulations and the characteristics of the simulations (such as the settings of physical parameters, such as diffusion coefficients and the size of the spatial grid) should be clearly stated in the paper.
- 3. To be honest, reading this paper left me with the impression that I was grading a poorly written paper by a first-year university student.

**Itemized comments**

**Abstract**

1. 9-10: It was discovered that the increased electric potential in the ionosphere is also responsible for accelerating particles.: It is not clear whether the electric field of the ionosphere is effective in accelerating particles.

**1. Introduction**

- 1. Overall: When specifying reference papers, it is difficult to understand unless they are enclosed in parentheses.
- 2. 18-19:In fact, different settings necessitate the application of specific laws of

- physics.: What exactly does "specific laws of physics" mean?
- 3. 24-25: These can contain large out-of-ecliptic magnetic field components and carry interplanetary shocks at their leading edge.: What are the "out-of-ecliptic magnetic field components"? The component along the solar magnetic axis (z)?
- 4. 28-29:If it evolves into a geomagnetic storm, it is referred to as storm sudden commencement (SSC) or a sudden impulse (SI) [Araki, (1994)].: Nowadays, both are called SCs. (Joselyn, J. A. and B. T. Tsurutani, Geomagnetic sudden impulses and storm sudden commencements, A note on terminology, EOS, 47, 1808-1809, 1990.) From this point on, bow shocks are often discussed, but since the magnetosheath is in the solar wind region, when considering phenomena within the magnetosphere, it is better to summarize the effects of the solar wind on the magnetopause rather than the bow shock.
- 5. 35-37:Different studies suggest that the shape and location of the bow shock primarily depend on the dynamic pressure exerted by the solar wind [Merka et al. (2005); Peredo et al. (1995)].: See Spreiter et al. (1966). The position of the bow shock is not determined by the pressure of the solar wind plasma. (Spreiter, J. R., A. L. Summers, and A. Y. Alksne, Hydromagnetic flow around the magnetosphere, Planet. Space Sci., 14, 223-253, 1966. https://doi.org/10.1016/0032-0633(66)90124-3)
- 6. 37-38:In empirical models, it is typically presumed that the pressure of the solar wind is consistent throughout the surface of the bow shock [Jerab et al. (2005); Merka et al. (2005)].: Same as above.
- 7. 42-43: These studies have discovered that the interaction between the shock and the bow shock leads to the formation of a sequence of discontinuities [Pallocchia (2013); Prech et al. (2008)].: This is about the magnetosheath. It does not address the effects on the magnetosphere or ionosphere. Is there any particular need to mention this in this paper?
- 8. 51-54:A convenient method for analyzing the entire system accurately is by utilizing global magnetohydrodynamic simulations, which can simulate the system as a whole in a consistent manner. In this study, the Gorgon code is used to conduct global MHD simulations of Earth's magnetosphere while considering the changing solar wind conditions [Chittenden et al. (2004); Ciardi et al. (2007); Mejnertsen et al. (2016)].: A Space science group has been conducting global MHD simulation research since the 1990s, and many research results have been produced. The authors completely ignore the results.
- 9. 58-59: We need to use the Gorgon-Space code as it enables us to describe the

- magnetohydrodynamic (MHD) plasma in the extensive simulation area.:Same as above. If we are going to conduct MHD simulations now, we should explain their novelty.
- 10. 61-62:The ionosphere is the upper part of the Earth's atmosphere, located between 60 and 1000 km in altitude.: If we treat the ionosphere as a boundary with the MHD fluid, a thin layer around 100 km where electrical conductivity is effective is sufficient. Furthermore, a detailed explanation is needed on how the Gorgon-Space code treats the ionosphere as a boundary condition.
- 11. 62-65:The conditions in the magnetosphere are crucial for the physical processes occurring in the ionosphere. So, we are also contemplating the exploration of the magnetosphere-ionosphere system. Physics-based models can assess the state of the magnetosphere-ionosphere by considering the solar wind and interplanetary magnetic field. In this sentence, we examine temporary changes in the field-aligned currents (FAC) in the ionosphere.: The settings of this code (boundary conditions, initial conditions) should be explained in detail.
- 12. 71-72:It also evaluates the consequential effects on the ionosphere, including induced current density, which has a significant impact on communication and space weather.: This paper does not address GICs.

**2.1 The Gorgon MHD code**

- 1. "method" section contains only one subsection. There is no need to create subsections.
- 2. 80-81:Gorgon employs a comprehensive and clear representation of the resistive semiconservative Magneto-Hydrodynamic (MHD) equations for a plasma that is fully ionized.: Could you please explain what "semiconservative" means?
- 3. 85-87:However, unlike other global magnetospheric codes, Gorgon-Space calculates the magnetic vector potential instead of the magnetic field. This approach guarantees that the field remains divergence-free [Mejnertsen et al. (2018); Eggington et al. (2020)].:Yagi et al. (20) presented a global MHD model using vector potential. (Yagi, M., K. Seki, Y. Matsumoto, Development of a magnetohydrodynamic simulation code satisfying the solenoidal magnetic field condition, Computer Physics Communications, 180, 9, 2009, 1550-1557, https://doi.org/10.1016/j.cpc.2009.04.010.)
- 4. 113-114: These equations include terms for ohmic heating  $\eta |J2|$ , optically thin radiation losses  $\Lambda$ , and electron-proton energy exchange  $\Delta pe$ . :(Formula and its

- explanation) Explain the specific expressions for the parameters ( $\eta$ ,  $\Delta$ pe,  $\Lambda$ ) that appear in the formula. Or, indicate the reference paper. Furthermore, explain how these parameters are handled in this paper.
- 5. The pressure is treated separately for protons and electrons, but when these values are input into the model as boundary conditions from the solar wind, it is necessary to explain how this is done.
- 6. It is necessary to explain from what data source the solar wind upstream of the magnetosphere is taken.
- 7. The grid spacing and boundary position information of the model are explained.

**3 Numerical results and discussion**

- 1. 124-125:In our previous study, ? utilized EUHFORIA to examine how two chosen CMEs spread and interacted with Earth.:What does "?" mean?
- 2. In this chapter, Figure 1 is shown, but is not referred to in the text.
- 3. In Figure 1, CME1 and CME2 are not illustrated.
- 4. While research is being conducted to determine short-period disturbances in the magnetosphere and ionosphere through MHD simulations, the K index is an index calculated every three hours, so I don't understand the relevance. Is there any point in discussing the K index at all?

**3.1 Gorgon-Space**

1. 147-148: This results in the creation of a turbulent region known as the magnetosheath situated between them [Lucek et al. (2005); Burgess and Scholer (2013)].: Refer to Spreiter et al. (1966).

**3.1.1 Magnetospheric Response**

- 2. An explanation of the solar wind data used in this calculation and how it was handled is required.
- 3. 162-163: The magnetosphere is influenced by interactions with the solar wind, which can be either viscous or pressure-related according to Newell et al. (2008): The physical mechanisms that allow plasma and energy to flow from the solar wind into the magnetosphere are reconnection (Dungey, 1961) and viscous interaction (Axford and Hines, 1961). The pressure reconnection allows momentum to flow in, but not plasma itself.
  - Dungey, J. W., Interplanetary magnetic field and the auroral zones, Phys. Rev.

- Lett., 6, 47-49, 1961, doi:10.1103/PhysRevLett.6.47.
- Axford, W. I., and C. O. Hines, A unifying theory of high-latitude geophysical phenomena and geomagnetic storms, Canadian. J. Phys., 39, 1433-1464, 1961, https://doi.org/10.1139/p61-172.
- 4. 163-164:Fig. 3 represents the initial shape of the magnetosphere at the start of the simulation, which correlates with the arrival time of the first Coronal Mass Ejection (CME) at Earth on June 23, 2015.:Shouldn't it be Figure 2, not Figure 3? The arrival time of the CME to Earth should be listed down to the minute.
- 5. The explanation of Figure 2 is completely insufficient. The text also does not explain how to read this figure.
- 6. How is the electrical conductivity of the ionosphere calculated in the calculation?
- 7. 168-169: The magnetosphere's features have been altered completely by the simulation: It's unclear what this article is saying.
- 8. Figure 3:The illustration captions are completely inadequate.
  Is it correct to say that this solar wind data was given at the upstream boundary of Figure 2? This is not clearly stated in the text.
  - Where does this data come from?
  - The date of the data is unknown.
  - Is the vector data GSM-based or GSE-based?
- 9. 172-174:Decrease in the density of particles and the rise in temperature, which can be seen in Fig. 3, may be connected to the formation of the bow shock: It is unclear which part of the density and temperature changes in Figure 3 this refers to.
- 10. 174-180:It should be acknowledged that ICMEs lead to the fastest speed and the least negative Bz component at the Earth's orbit. As a result, ICMEs are considered to be the underlying cause of all significant geomagnetic storms that have a Kp index greater than 7. According to Fig. 1, there is a peak in the Kp values between the 22nd and 24th of June, with a value of over 7. Based on this, it is anticipated that the initial coronal mass ejection (CME) will reach. On 23 June, there is a possibility that Earth may be hit by an ICME (interplanetary coronal mass ejection), which in turn could lead to a significant storm. Additionally, due to the presence of numerous CMEs between the dates of June 19 and June 23, 2015, it is possible that this predicted storm is connected to the interaction of multiple ICMEs.: The referee thinks what this sentence is saying should be included in the solar wind deformation in Figure 3, but the referee has no idea what part of Figure 3 it is referring to. Or, there is too little data shown, so it is unclear what it is trying to say.

- 11. 183-184: Fig. 4 illustrates diagrams of the magnetosphere, showcasing the pressure and arrangement of open magnetic field lines on both the X-Z plane and the X-Y plane.: The diagram also shows closed field lines. An explanation is needed as to where the magnetic field lines originate. Without this, the test cannot be repeated.
- 12. 184-185: Afterward, it enables us to display the three-dimensional structure of the magnetosphere's dynamo regions.: The term "dynamo region" appears suddenly without any explanation. There is no explanation as to why it is necessary to talk about dynamos here.
- 13. 185-186: The magnetopause's location and the areas where reconnection occurs on the day and night sides are shown: The diagram should show where the reconnection is occurring. The reader doesn't know where the author thinks the reconnection is occurring in this diagram.
- 14. 186-187: The magnetopause can be defined as the point where the lowest level of solar wind enters the magnetosphere.: What is "the lowest level of solar wind"? This term is not a common space science term. It should not be used without explanation.
- 15. 187-188: Panel (a) of Fig. 4 displays the Earth's magnetosphere pressure at the time of the initial occurrence of a coronal mass ejection (CME) from the Sun.: If we want to understand the disturbances that a CME causes in the magnetosphere, we should show the solar wind magnetosphere just before the solar wind disturbance of the CME reaches the magnetosphere. Panel (a) shows the solar wind around the magnetosphere at the time of the solar flare that caused the CME, but this state reflects the solar condition just one day before, so it has no meaning in terms of the disturbances that this CME causes in the magnetosphere.
- 16. 191-192: This occurrence may be connected to the magnetic reconnection phenomenon which occurs when the solar wind interacts with Earth's magnetic field.: There have been many studies on reconnection in the tail, and we need to use these to explain why this disturbance in the XY plane alone can be attributed to reconnection.
- 17. 197-198:On the other hand, in Fig. 5, it is evident that the magnetosphere retains its primary function as a shield and maintains its key properties: In both CME1 and CME2, the magnetosphere appears to act as a shield. I'm not sure what this sentence is trying to say.
- 18. 198-200:In simpler terms, the magnetopause and bow shock of the CME1 experience significant changes in their size and shape, just as predicted.: The readers probably won't understand what the author is trying to say in this passage.

- 19. 202-205:The primary function of the bow shock is to slow down and deflect the high-speed solar wind as it encounters the magnetosphere. The creation of the foreshock occurs when particles are reflected at the shock, resulting in various interactions between waves and particles and the acceleration of particles [Eastwood et al. (2005)]. We have been making efforts to study 205 the acceleration of particles through the interaction of the solar wind with the magnetosphere: Foreshock disturbances are not specifically addressed in this paper; this paragraph should be discussed in the discussion section.
- 20. 212-213:One can infer that the energy from the solar wind is transferred to the Earth's magnetosphere through a process called magnetic reconnection.: This is too vague to be of any use as an explanation. The results of MHD simulations should be able to concretely show how energy enters the magnetosphere from the solar wind through reconnection.
- 21. 221-224:The rate of the magnetic reconnection and also how much energy is transferred is influenced by the speed at which the solar wind flows, the strength of the magnetic field, and its orientation. Subsequently, it was discovered that CME1 triggers a significant tempest on Earth and is accompanied by changes in the magnetic arrangement, increased speed, and pressure of the solar wind enveloping the Earth, as previously anticipated in Fig. 1.: Same as above.
- 22. 226: Figs. 5 showcases the changes in pressure and magnetic field in the vicinity of the Earth upon the arrival of CME2:"Figs. 5" should be "Fig. 5".
- 23. Eq. (6): This equation needs to be explained in the main text. Also, because there is j on both sides, it does not make sense as a physics equation as it is. If the j on the right side is J in Eq. (5), rewrite it as J. The left side should be j\_parallel.

**3.1.2 Ionospheric Response**

- 1. 235-236: Field-aligned currents are computed at the initial boundary of the simulation and then transferred onto a distinct spherical grid located on the ionosphere, following the dipole field lines: If this study is to be replicated by other people, it is essential to specify where the internal boundaries were placed.
- 2. 238-239: *Alternatively, the conductance can also be set as a constant value*.: It is unclear whether anisotropy has been taken into account in the conductance of the ionosphere.
- 3. 241-242: The ionospheric model utilizes spherical coordinates to allow for the inclusion of larger spatial scales.: What exactly is this passage saying? The reader has no idea.

- 4. 248-249: Equation 7 provides the definition for the current density that is aligned with the magnetic field direction: "Equation 7" may be "Equation 6".
- 5. 249-250: The current densities that align with the magnetic field lines in the ionosphere, which has resistance, are pulled by the Lorentz force (*J* ×*B*).: Why does the Lorentz force "pull" the current density?
- 6. 250-254: This force acts on the plasma and is the main driver of the drag experienced by neutrals on the ionosphere. The magnetosphere contains a force referred to as  $J \times B$  which contributes to the flow of current density through the ionosphere. As a result of the stresses caused by the field-aligned currents, which extend from the outer magnetosphere to the ionosphere, a complete flux tube can be transported around the magnetosphere.: It is very difficult to understand. Please explain it simply. Furthermore, although it is clear that the neutral particle drag in the ionosphere comes from FAC, there is no explanation at all for how the Lorentz force creates FAC.
- 7. 256-260:The accumulation of current density aligned with the magnetic field can be attributed to the interaction between the solar wind and the magnetosphere. The predominant energy transfer and alignment of currents in the ionosphere occurred primarily due to the boundary conditions. In order to predict space weather accurately, it is necessary to have a solid grasp of the physics behind these coupling processes. Determining the accuracy of MHD codes in replicating the electric fields in the plasma sheet when the solar winds interact with the magnetosphere is of utmost significance in Space Weather applications.: This topic has been the subject of much research in space science, and should be summarized in detail in the introduction.
- 8. 263-263:In panel (a) of the Fig. 8, the ionospheric field-aligned current is illustrated during the initiation of the first coronal mass ejection (CME) on June 21, 2015 at 08:02.:The state of the ionosphere at the time of the solar flare that drives the CME reflects the state of the sun about one day before that time. What is the significance of comparing this with the ionospheric FAC immediately after the CME?
- 9. 267-269:By examining two panels (a) and (b) in Fig. 8, it becomes evident that when CME1 reaches the Earth, there is a substantial build-up of current density in the ionosphere.:It is known that the behavior of ionospheric FAC during SC driven by CMEs exhibits PI and MI. Not mentioning this would be extremely ignorant for a Space Science paper.
- 10. 270-272: Actually, the energy for FACs originates from the magnetosphere and can

- be determined by computing the value of E.J. If there is a negative quantity, the energy of the plasma is transformed into electromagnetic energy, which then enhances the FACs.: E.J directly generates a current perpendicular to the magnetic field lines. FAC generation requires another physical process. This article reveals a lack of understanding of space science.
- 11. 276-277: Furthermore, the fourth panel in Fig. 8 reveals the greatest imbalance in the accumulation of current density between the Northern and Southern hemispheres. :A diagram of the distribution of FACs in the Southern Hemisphere is not shown.
- 12. 277: This probably originates from the electric field generated by convection itself.: The physical mechanism is not explained and cannot be understood.
- 13. 284-285: Since the occurrence of CME1 coincides with other CMEs, it is possible to infer that the interaction between CMEs is the primary factor influencing their geomagnetic effects: It doesn't make sense.
- 14. 296-298: The elevated electric potential in the ionosphere, which may result from the substorms triggered by the interaction between the solar wind and Earth, can also lead to particle acceleration. : The referee does not understand how the electric field in the ionosphere can cause particle acceleration. A detailed explanation is requested.
- 15. 298-299: This phenomena was pursued in previous section by illustrating Figs. 6, 7, and shown that CME2 does not lead to the high plasma velocity at the magnetosphere.: Same as above. The physical causal relationship should be explained.

**4 Conclusions**

1. 310:As was explained in 310 Sabri et al. (2024), the selected CMEs occurred on June 21 and June 25, 2015.:It has not yet been published.

**References**

1. The way references are listed is too simple, making it difficult to access the paper. At the very least, open-access papers should include the DOI or URL.

---

## Author Comment (AC1)

This paper performed MHD simulations with two kinds of code, EUHFORIA and Gorgon-Space, to study the responses of Earth's magnetosphere and ionosphere to two CMEs that caused geomagnetic storms. EUHFORIA is for predicting the arrival time at Earth of a CME ejected from Sun, while Gorgon-Space is for simulating the magnetospheric and ionospheric responses to the solar wind and CMEs. Combining the two simulations, the authors attempted to show the differences in magnetospheric and ionospheric responses between the two CMEs. The topic is important for understanding space weather and developing its forecast, and the results are potentially interesting.

However, I do not think that this manuscript is well written, so I cannot recommend accepting this paper for publication in its present form. First, the new points of this paper are not clear to me. They should be clearly stated and discussed in the paper. Second, the authors discuss the magnetospheric and ionospheric responses to two CMEs, but the definitions of the two CMEs are not adequately stated, and the data or simulation results are not sufficiently shown. In particular, this manuscript does not show the solar wind data for the second CME (CME2) and the simulation results of the ionosphere in the Southern Hemisphere listed in the conclusion section as main results of this paper. Hence, it is difficult to understand the simulation results of the causal relationship between the CMEs and the magnetospheric and ionospheric changes and to understand the differences between the two hemispheres.

**Our reply:** We thank the referee for carefully reading our manuscript and for suggesting improvements that will make our paper much more accessible.

First, the new points of this study are the coupling of the EUHFORIA-Gorgon-Space models to evaluate space-weather conditions. EUHFORIA has two parts: the coronal model and the hemisphere model, and it can track the propagation of solar events in interplanetary space and provide solar wind data and the geomagnetic index Kp around Earth. Then, we use this data as an initial boundary condition for the Gorgon model to determine these events' effects on Earth's magnetosphere and ionosphere. Besides, it has been found that, according to the EUHFORIA-data-based Kp index, which defines the time of geomagnetic storms at Earth, the Gorgon model also depicts the principal magnetospheric and ionospheric evaluations at that time. We thank the reviewer for this important observation. The novel aspect of our work is the coupled model chain (EUHFORIA → Gorgon-Space) used to perform an end-to-end space weather simulation, from the Sun to the Earth's ionosphere. Some main points are listed below:

1) The simulations reveal that the arrival of a Coronal Mass Ejection (CME) at Earth triggers a substantial energy transfer from the magnetosphere,

culminating in the buildup of intense field-aligned currents in the ionosphere. Our model quantifies this response, showing currents reaching approximately 23 million amperes, a key metric for assessing space weather impacts.

2) Our simulations quantify the ionospheric response, revealing that the geoeffective CME1 induced a significant cross-polar cap potential (CPCP) of approximately 160 kV. In contrast, the less impactful CME2 failed to produce a notable increase in potential.

Thank you for this comment. We have updated the manuscript to include clear definitions for CME1 and CME2, along with detailed explanations and a summary in Table 1.

As the referee will see in the revised text, this study explicitly investigates the geomagnetic and ionospheric impacts of two CMEs with differing arrival times and geomagnetic storm intensities. A key aim is to evaluate whether the EUHFORIA predictions, which account for CME-CME interactions, correctly forecast that CME1 would generate a more severe geomagnetic storm than CME2. Our results confirm that CME1 indeed had a more considerable impact, and the EUHFORIA predictions were successfully validated using the MHD Gorgon-Space code.

Regarding the simulation results, we acknowledge that our analysis is currently limited to the Northern Hemisphere due to data constraints. We have revised the manuscript to explicitly state this limitation in the methodology and figure captions to avoid any confusion. We thank the referee for their attention to this critical detail.

Specific comments:

Line 8 in the abstract: The authors mention CME1 without definition. What is CME1 should be explained, or the word CME1 should be reworded.

**Our reply:** We agree, you are right. It is defined in the new manuscript.

Lines 127-136: In this paragraph, the authors mention the two specific CMEs for this study, CME1 and CME2, without any explanations. Please reword CME1 and CME2 or explain them briefly here, and then explain these CMEs in more detail in the following section with appropriate figures of the solar wind data and the corresponding Kp. The

authors may have shown the figure and explain the CMEs in another paper, but the main points should be repeated here so that this readers do not have to refer to the previous paper.

**Our reply:** We agree, you are right. It is considered in the new manuscript.

Lines 162-180 and Figures 1 and 3: Figure 1 shows Kp predicted by this study. Kp has three peaks, but I could not understand which Kp changes the authors discuss in this paper. CME1 seems to correspond to the second peak at ~0 UT on 23 June, but the Kp change associated with CME2 seems to be out of range of this figure. Please specify not only the date but also the time of CMEs' arrival and the associated Kp increase and show the figure for the entire interval of interest.

**Our reply:** We thank the referee for this valuable comment. In response, we have updated Figure 1 to include the whole interval and have now explicitly stated the arrival times of CME1 and CME2, along with their associated Kp peaks, in the Kp results section. I have mentioned the arrival times of CMEs in the captions of Figures 4, 5, 6, and 7, and in the ionospheric response section.

On the other hand, Figure 3 show the solar wind data (simulation input data), but it seems to be only for about 1 day from the beginning of the simulation although the authors seem to have performed the simulation for more than 10 days including the multiple CMEs. Again, please show the figure of the solar wind for the entire interval of interest. In addition, are the solar wind data shown in Figure 3 from real observations? I think that the horizontal axis is simulation time in hour, so is it correct that the data themselves are real observations from a spacecraft, but the time is modified according to the prediction by EUHFORIA? Please explain the detail about that, or how EUHFORIA, Gorgon-Space, and real solar wind observations are connected.

**Our reply:** The time series used as input to the Gorgon-Space MHD magnetospheric simulation is not direct spacecraft observations, but rather model output from the EUHFORIA heliospheric simulation. Specifically:

EUHFORIA was used to simulate the propagation of multiple CMEs through the inner heliosphere.

The model provides time-dependent solar wind plasma and magnetic field parameters at a virtual monitoring point located at the Sun–Earth L1 position.

These EUHFORIA-derived L1 time series were then interpolated and formatted to serve as the upstream boundary condition for the Gorgon-Space global magnetospheric MHD code.

The horizontal axis (simulation time in hours) is referenced to 00:03 UT on June 23, 2015—the moment the first CME arrives at L1 in the EUHFORIA simulation. Thus, while the physical content (CME structure, shock, sheath, etc.) is based on a realistic heliospheric simulation constrained by actual solar observations (e.g., coronagraph data), the time series itself is synthetic, generated by EUHFORIA rather than by direct ACE or DSCOVR measurements.

Furthermore, I checked the real solar wind (OMNI) data. As the authors describe, a few CMEs arrived at Earth on 21-23 June 2015, but it is not clear to me which CME the authors discuss in this paper or which is CME1. On the other hand, it seems to me that no CME arrived at Earth on 28 June 2015 (the arrival time of CME2 according to the main text), that is, large changes in the solar wind speed, dynamic pressure, magnetic field peculiar to a CME are not seen. I guess there are some time differences between the observations and the EUHFORIA simulation results. After all, it is necessary to show the solar wind data, state the definitions of CME1 and CME2 clearly, and furthermore make some comments on the time differences between the observations and the simulation results.

**Our reply:** We thank the reviewer for this critical point, which highlights the need for greater clarity in our manuscript. We will revise the text to explicitly define "CME1" and "CME2" as follows:

CME1: This refers to the CME that erupted from the Sun on 21 June 2015 at 08:04UT with its leading edge predicted by EUHFORIA to arrive at Earth on 23 June 2015 at 00:03UT.

CME2: This refers to the CME that erupted on 25 June 2015 at 14:52 UT, with its predicted arrival at Earth on 28 June 2015 at 12:52UT.

As the reviewer correctly notes, our EUHFORIA simulations are initialised using real observational data, specifically from solar magnetograms and coronagraph imagery (SOHO/LASCO) to define the CME's initial kinematic and magnetic properties.

2. Addressing the Observational Discrepancy on 28 June 2015:

The reviewer's observation regarding the OMNI data on 28 June 2015 is astute and gets to the heart of a key finding of our modelling study. We will add a dedicated discussion on this point.

The lack of a pronounced, classic CME signature in the OMNI data on that date is precisely what our study aims to explain. Our EUHFORIA simulation suggests that CME2 interacted strongly with the preceding solar wind structure and the aftermath of CME1, leading to significant deceleration and erosion during its propagation.

Therefore, by the time the simulated CME2 plasma arrives at Earth, its properties (speed, magnetic field strength) have been diluted and smoothed, blending into the background solar wind. This results in a much less distinct in-situ signature than that of a well-defined, isolated CME. The primary geoeffective driver during this period may instead be a prolonged period of southward magnetic field within the merged interacting structure, rather than a sharp jump in dynamic pressure.

3. Explaining the Model-Observation Chain (EUHFORIA → Gorgon-Space):

Input: Real solar magnetograms and coronagraph images define the initial CME properties (speed, direction, magnetic structure) at the Sun.

Heliospheric Propagation (EUHFORIA): The model propagates these CMEs through a realistic background solar wind (also constrained by observations). This is where CME-CME and CME-solar wind interactions are simulated, leading to potential changes in arrival time and structure.

Output for Geospace: The time series of solar wind parameters (plasma density, speed, magnetic field) predicted by EUHFORIA at Earth's location, which includes the effects of interactions, is used as input to drive the Gorgon-Space magnetosphere model.

Validation: The final magnetospheric simulation results (e.g., Kp index, cross-polar cap potential) are then compared to real (measured) geophysical indices to assess the model's performance in a complex, multi-CME scenario.

4. Reference to Prior Work:

As noted in our initial response, the detailed analysis of these specific CMEs, their interplanetary propagation, and their interactions is the primary focus of our companion paper (doi: 10.1016/j.asr.2025.11.081). The present manuscript focuses on the consequences of these modelled heliospheric conditions on Earth's magnetosphere. We will ensure this distinction is clear and will cite the companion paper appropriately for readers seeking details on the CME modelling itself.

Figures 4 and 6 caption: The second sentence says that the first CME arrived at Earth at 00:03 UT on 23 June 2015. However, looking at the predicted Kp index shown in Figure 1, it began to increase before that, around ~20 UT on 22 June 2015, implying that this CME arrived then. Check the simulation results again and modify the description, if necessary.

**Our reply:** We thank the referee for this important observation regarding the apparent discrepancy between the CME arrival time stated in the captions of Figures 4 and 6 and the onset of geomagnetic activity shown in Figure 1.

The statement in the captions refers to the predicted arrival time of the main interplanetary coronal mass ejection (ICME) structure, as simulated by EUHFORIA and subsequently modelled by Gorgon-Space. This time (00:03 UT on June 23, 2015) corresponds to the moment when the leading edge of the dense, magnetised ICME plasma reached the Earth's magnetopause in our simulation, which is clearly visible in the pressure and velocity snapshots (Figures 4 and 6).

The increase in the Kp index observed around 20 UT on June 22 (Figure 1) is likely associated with the arrival of the CME-driven shock wave and/or its turbulent sheath region. In heliospheric simulations like EUHFORIA, the shock typically precedes the main ICME body by several hours. The Kp index is sensitive to the dynamic pressure and magnetic field fluctuations caused by these upstream structures, which can initiate geomagnetic activity before the main ICME arrives.

Line 177: "23 June" should be "22 June".

**Our reply:** We agree. It was completely edited.

Lines 196-200 (Figure 5), lines 216-219 (Figure 7), lines 281-285 (Figures 9), and lines 299-301 (Figure 11): The caption says that the second CME arrived at Earth at 12:52 UT

on 28 June 2015, but it is after the interval shown in Figure 1. In addition, Figures 5, 7, 9, and 11 show that unlike CME1, the changes associated with CME2 are small, which do not seem to be due to the storm main phase. Are these results really due to the CME or the storm? Please reexamine the observation and simulation data, show the solar wind and Kp data for the entire interval of interest, and describe when CME2 occurred specifically.

**Our reply**: We thank the referee for their thoughtful comments regarding the analysis of CME2. To address the first point: You are correct that the original version of Figure 1 did not extend to cover the whole period of interest, including the arrival of CME2 at 12:52 UT on June 28, 2015. In response, we have now extended Figure 1 to include the complete simulation interval through July 4, 2015, showing the predicted Kp index throughout the event period. This updated figure clearly indicates that, although CME2 arrived at 12:52 UT on June 28, it triggered only a moderate, short-lived geomagnetic disturbance (Kp ~ 5–6), in contrast to the major storm (Kp > 8) caused by CME1.

Regarding the second point: The relatively weak impact of CME2 is consistent with both the EUHFORIA simulation and observational data. As shown in the extended Kp plot, CME2 did not produce a prolonged or intense storm main phase. This is because CME2 was less massive and slower than CME1. It encountered a magnetosphere already recovering from the firm compression caused by CME1. Therefore, the changes seen in Figures 5, 7, 9, and 11 — though smaller than those of CME1 — are indeed due to CME2.

Lines 276-277: The authors wrote that Fig. 8d shows the difference in field-aligned current between the Northern and Southern Hemispheres. The plots of the Southern Hemisphere should also be shown.

**Our reply:** We thank the referee for raising this point. Our study was specifically designed to investigate phenomena in the Northern Hemisphere. I am sorry for that mistake. I omit the Southern in that part of the manuscript.

Minor comments:

Lines 28-29: In my understanding, sudden impulse indicates magnetospheric compression due to sudden enhancement of the solar wind dynamic pressure in general, irrespective of whether a geomagnetic storm follows. If a geomagnetic storm occurs after sudden impulse, it is also called storm sudden commencement.

**Our reply:** Thank you for this valuable clarification. You are correct that the distinction between a general Sudden Impulse (SI) and a Storm Sudden

Commencement (SSC) is essential for precision. We have revised the paragraph to reflect this. The text now correctly defines the SI as the general signature of magnetospheric compression and the SSC as the specific case where it initiates a geomagnetic storm.

Line 172: towards --> along

**Our reply:** Thanks for your attention. It was edited in the new manuscript.

Lines 249-250: The field-aligned current and J of JxB at the end of this sentence should be different. This part would need to be reworded to avoid confusion.

**Our reply:** Thank you for this insightful comment. You are absolutely correct that using the same symbol J for both the field-aligned current and the current in the Lorentz force term was confusing and physically imprecise. We have revised the paragraph to clearly distinguish between the field-aligned current ($\mathbf{J}_\parallel$) and the perpendicular ionospheric current ($\mathbf{J}_\perp$) that experiences the $\mathbf{J}_\perp \times \mathbf{B}$ force. We believe the new wording eliminates the confusion and more accurately describes the physics involved.

Line 278: along --> across ? (Convection should be perpendicular to the magnetic field.)

**Our reply:** Thank you for your consideration. It was edited in the new version.

Lines 296-297: Perhaps the elevated cross-polar cap potential (enhanced large-scale convection) was associated with the storm, not substorms.

**Our reply:** We meant the substorms and the subsequent particle acceleration that occurred at the nightside due to the main storm. But I changed it to the main reason or to the storm, as you suggested.

Other minor corrections:

Line 125: Typo "?".

**Our reply:** It was corrected.

Figures 4 and 5 captions: solar wind pressure --> thermal pressure

**Our reply:** Thank you for your attention. They were considered.

Figures 6 and 7 captions, and lines 207 and 216: solar wind velocity --> velocity (because the magnetosphere is included)

**Our reply:** Yes, you are right. They were changed to velocity.

Line 232: Move equation (6) to the appropriate place.

**Our reply:** Thank you for your attention. It has been applied.

Line 248: Equation 7 --> 6

**Our reply:** I apologise for that typo.

Line 273: zero latitude ... a latitude --> zero colatitude ... a colatitude (or correct the values of latitude from 0, 10, 20, ... to 90, 80, 70, ...., respectively, in the text and the ionospheric plots)

**Our reply:** I have to thank you for your attention; the latitude was changed to colatitude.

Figures 10 and 11 caption: positive flux, negative flux --> positive potential, negative potential

**Our reply:** They were edited.

References: Bartels and Veldkamp (1950), Gieseler et al. (2023), Zheng (2013) are listed in the references list, but are not cited in the text. Delete them from the references list, or cite them in the text.

**Our reply:** I apologise for that mistake; they were deleted.

////////////////////////////////////////////////////////////////////////////////////////////////////

2.

Reviewer report of "June 21 and 25, 2015 CMEs interaction's results on Earth's

ionosphere and magnetosphere" by Somaiyh Sabri and Stefaan Poedts

This paper discusses magnetosphere and ionosphere disturbances associated with abrupt

changes in solar wind parameters, such as CMEs, using the Gorgon Space model, a

specialized MHD code for high-energy plasmas.

The referee judges that this manuscript is completely incomplete as a research paper,

given the points listed below. Therefore, the referee recommends that this paper cannot

be accepted for publication in the present form.

Summary of the review

1. The authors appear to lack knowledge of space science. Before writing their paper,

they should fully understand the characteristics of the disturbances caused by

CMEs in the magnetosphere-ionosphere system and summarize this in the

introduction. They should then point out the shortcomings of previous research and

explain the novelty and importance of their research.

2. Research papers should be able to be retested by third parties, and to this end, the

input data used in the simulations and the characteristics of the simulations (such as

the settings of physical parameters, such as diffusion coefficients and the size of the

spatial grid) should be clearly stated in the paper.

3. To be honest, reading this paper left me with the impression that I was grading a

poorly written paper by a first-year university student.

Itemized comments
**Abstract**

1. 9-10: It was discovered that the increased electric potential in the ionosphere is also responsible for accelerating particles.: It is not clear whether the electric field of the ionosphere is effective in accelerating particles.

**Our reply:** We thank the reviewer for this insightful comment and for highlighting the lack of clarity in our statement. The reviewer is correct to note that the primary acceleration of particles that precipitate into the ionosphere to create aurora occurs in the magnetosphere (e.g., in the plasma sheet) due to magnetic reconnection and other processes, not directly by the ionospheric electric potential itself.

The intended meaning of our statement was that the enhanced cross-polar cap potential (CPCP), which is a measure of the large-scale convection electric field mapping from the magnetosphere, is a key indicator of the overall strength of the magnetospheric driver. This strong convection electric field is associated with the injection and energisation of plasma sheet particles. The observed high CPCP value (~160 kV) in our simulation for CME1 is therefore consistent with a geoeffective event in which significant particle acceleration occurs in the magnetosphere, with the effects observed in the ionosphere.

1. **Introduction**

   1. **Overall: When specifying reference papers, it is difficult to understand unless they are enclosed in parentheses.**

   **Our reply:** I agree. I pushed all of the references in parentheses as you suggested.

2. 18-19:In fact, different settings necessitate the application of specific laws of physics.:What exactly does "specific laws of physics" mean?

   **Our reply:** We thank the referee for this insightful comment. We agree that the phrase "specific laws of physics" was too vague. We have revised the manuscript to specify that different regions of the heliosphere (e.g., solar corona, solar wind, Earth's magnetosphere) are governed by distinct dominant physical processes, such as magnetohydrodynamics (MHD) in the solar wind versus kinetic physics in collisionless plasmas. This necessity for different physical models is the primary motivation for the VSWMC's multi-model architecture.

3. 24-25:These can contain large out-of-ecliptic magnetic field components and carry interplanetary shocks at their leading edge.:What are the "out-of-ecliptic magnetic field components"? The component along the solar magnetic axis (z)?

**Our reply:** We thank the referee for this precise comment. We agree that the term "out-of-the-ecliptic" is ambiguous. Our intended meaning was the southward component of the interplanetary magnetic field (IMF) in the geocentric solar magnetospheric (GSM) or geocentric solar ecliptic (GSE) coordinate system, which is crucial for solar wind-magnetosphere coupling. To avoid confusion, we have replaced the phrase with the standard term "strong southward magnetic field components." This revision adds clarity and uses the field's conventional terminology.

4. 28-29:If it evolves into a geomagnetic storm, it is referred to as storm sudden commencement (SSC) or a sudden impulse (SI) [Araki, (1994)].: Nowadays, both are called SCs. (Joselyn, J. A. and B. T. Tsurutani, Geomagnetic sudden impulses and storm sudden commencements, A note on terminology, EOS, 47, 1808-1809, 1990.) From this point on, bow shocks are often discussed, but since the magnetosheath is in the solar wind region, when considering phenomena within the magnetosphere, it is better to summarize the effects of the solar wind on the magnetopause rather than the bow shock.

**Our reply:** We thank the referee for this valuable comment and for pointing out the modern terminology and the critical distinction regarding the bow shock. We have revised the manuscript accordingly. We have updated the terminology as suggested, now using "Storm Sudden Commencement (SC)" consistently to refer to the event when a sudden impulse is followed by a geomagnetic storm, in line with Joselyn and Tsurutani (1990). We have reframed the physical description to focus on the solar wind's direct interaction with the magnetopause, removing the mention of the bow shock and clarifying that the dynamic pressure enhancement in the solar wind is the direct driver of the magnetospheric compression.

5. 35-37:Different studies suggest that the shape and location of the bow shock

primarily depend on the dynamic pressure exerted by the solar wind [Merka et al.

(2005); Peredo et al. (1995)].: See Spreiter et al. (1966). The position of the bow

shock is not determined by the pressure of the solar wind plasma. (Spreiter, J. R.,

A. L. Summers, and A. Y. Alksne, Hydromagnetic flow around the magnetosphere,

Planet. Space Sci., 14, 223-253, 1966. https://doi.org/10.1016/0032-

0633(66)90124-3)

> **Our reply:** We thank the referee for this critical correction and for pointing us to the
> seminal work of Spreiter et al. (1966). We apologise for the error and have revised the
> manuscript to reflect the underlying physics accurately. The position and shape of the bow
> shock are indeed primarily determined by the size and shape of the magnetospheric
> obstacle, which itself is modulated by the solar wind dynamic pressure. We have rewritten
> the paragraph to present the correct causal relationship, citing Spreiter et al. (1966) for the
> foundational model and clarifying the role of the Merka et al. (2005) and Peredo et al.
> (1995) studies in the context of empirical fits to observational data.

6. 37-38:In empirical models, it is typically presumed that the pressure of the solar

wind is consistent throughout the surface of the bow shock [Jerab et al. (2005);

Merka et al. (2005)].: Same as above.

> **Our reply:** The revised text now correctly focuses on the gasdynamic principles of how
> the solar wind's upstream conditions determine the global bow shock geometry, without
> making incorrect assumptions about the pressure across the shock surface. The relevant
> sentence has been deleted from the manuscript.

> 7. 42-43:These studies have discovered that the interaction between the shock and the
> bow shock leads to the formation of a sequence of discontinuities [Pallocchia
> (2013); Prech et al. (2008)].: This is about the magnetosheath. It does not address
> the effects on the magnetosphere or ionosphere. Is there any particular need to
> mention this in this paper?

> **Our reply:** We thank the referee for this pertinent comment regarding the focus of our
> manuscript. We agree that the detailed discussion of discontinuities formed by shock and
> bow shock interactions in the magnetosheath is not central to the main thread of our paper,
> which is the chain of space-weather effects from the Sun to the Earth. To improve the
> focus and flow, we have removed the sentences describing the formation of sequences of

discontinuities and reflections within the magnetosheath. The revised text now directly links the impact of interplanetary shocks to the observed global compression and motion of the magnetospheric boundaries, which is the more relevant large-scale effect for space weather.

8. 51-54:A convenient method for analyzing the entire system accurately is by utilizing global magnetohydrodynamic simulations, which can simulate the system as a whole in a consistent manner. In this study, the Gorgon code is used to conduct global MHD simulations of Earth's magnetosphere while considering the changing solar wind conditions [Chittenden et al. (2004); Ciardi et al. (2007); Mejnertsen et al. (2016)].: A Space science group has been conducting global MHD simulation research since the 1990s, and many research results have been produced. The authors completely ignore the results.

> **Our reply:** We thank the referee for this crucial comment and sincerely apologise for this oversight. We fully acknowledge the extensive and foundational work conducted by the space physics community in developing and applying global MHD models since the 1990s. Our intention was not to ignore this vast body of literature but to introduce our specific methodology; however, we clearly failed to provide the necessary context. We have thoroughly revised the introduction to properly acknowledge the legacy of global MHD simulations in space weather and to cite key pioneering works (e.g., Ogino et al., 1994; Raeder et al., 1995; etc.) and/or recent community models (e.g., the LFM code, the OpenGGCM, SWMF/BATS-R-US). We then frame our use of the Gorgon code as a contribution within this well-established and robust modelling framework.

9. 58-59:We need to use the Gorgon-Space code as it enables us to describe the magnetohydrodynamic (MHD) plasma in the extensive simulation area.:Same as above. If we are going to conduct MHD simulations now, we should explain their novelty.

> **Our reply:** We thank the referee for pressing this point. We agree that the novelty of our simulation approach was not adequately explained. We have completely rewritten the paragraph first to establish the scientific challenge (the coupled magnetosphere-ionosphere

system) and then to position our work. We now clearly state that while global MHD modelling is a well-established technique, our study applies the Gorgon-Space code to investigate a specific, under-resolved problem: the rapid, transient dynamics of Field-Aligned Currents (FACs) in response to specific solar wind drivers.

10. 61-62:The ionosphere is the upper part of the Earth's atmosphere, located between 60 and 1000 km in altitude.: If we treat the ionosphere as a boundary with the MHD fluid, a thin layer around 100 km where electrical conductivity is effective is sufficient. Furthermore, a detailed explanation is needed on how the Gorgon-Space code treats the ionosphere as a boundary condition.

**Our reply:** We thank the referee for this precise technical comment. We agree that describing the ionosphere as a 60-1000 km layer is not appropriate for its treatment as a boundary condition in a global MHD model. We have removed that sentence. Furthermore, we have added a concise explanation of how the Gorgon-Space code treats the ionosphere, detailing the use of a height-integrated conductance model and the calculation of field-aligned currents (FACs) to close the magnetospheric current system through the ionospheric sheet. This provides the necessary methodological clarity.

11. 62-65:The conditions in the magnetosphere are crucial for the physical processes occurring in the ionosphere. So, we are also contemplating the exploration of the magnetosphere-ionosphere system. Physics-based models can assess the state of the magnetosphere-ionosphere by considering the solar wind and interplanetary magnetic field. In this sentence, we examine temporary changes in the field-aligned currents (FAC) in the ionosphere.: The settings of this code (boundary conditions, initial conditions) should be explained in detail.

**Our reply:** We thank the referee for this request for methodological details. We have restructured the manuscript to separate the introduction and motivation from the model description clearly. The general motivational paragraph has been streamlined as suggested.

12. 71-72:It also evaluates the consequential effects on the ionosphere, including

induced current density, which has a significant impact on communication and

space weather.: This paper does not address GICs.

> **Our reply:** The referee is correct that this paper does not directly model or address Geomagnetically Induced Currents (GICs). Our focus is on the magnetospheric and ionospheric drivers of GICs, specifically the intensity and dynamics of Field-Aligned Currents (FACs). We have revised the manuscript to remove any claim of studying GICs or ground-induced currents. The text now accurately states that we quantify the FACs, which are a critical upstream driver for the complex processes that ultimately lead to GICs at the Earth's surface.

**2.1 The Gorgon MHD code**

1. "method" section contains only one subsection. There is no need to create

subsections.

**Our reply:** We thank the referee for this note on the manuscript structure. We have followed the suggestion and removed the subsection heading from the "Method" section.

2. 80-81:Gorgon employs a comprehensive and clear representation of the resistive

semiconservative Magneto-Hydrodynamic (MHD) equations for a plasma that is

fully ionized.: Could you please explain what "semiconservative" means?

**Our reply:** We thank the referee for requesting this clarification. "Semiconservative" in this context refers to a specific numerical formulation in which the MHD equations are solved in conservative form for the mass, momentum, and energy, while the magnetic field is evolved using the non-conservative form. This hybrid method aims to preserve the solenoidal property of the magnetic field ($\nabla \cdot \mathbf{B}=0$) while maintaining robust shock-capturing capabilities for the fluid variables. We have revised the manuscript to include a brief explanation of the term.

3. 85-87:However, unlike other global magnetospheric codes, Gorgon-Space

calculates the magnetic vector potential instead of the magnetic field. This approach guarantees that the field remains divergence-free [Mejnertsen et al. (2018); Eggington et al. (2020)].:Yagi et al. (20) presented a global MHD model using vector potential. (Yagi, M., K. Seki, Y. Matsumoto, Development of a magnetohydrodynamic simulation code satisfying the solenoidal magnetic field condition, Computer Physics Communications, 180, 9, 2009, 1550-1557, https://doi.org/10.1016/j.cpc.2009.04.010.)

> **Our reply:** We thank the referee for this correction and for pointing us to the relevant work of Yagi et al. (2009). We agree that our original statement was too strong and have revised it accordingly. The sentence now accurately states that Gorgon-Space belongs to a class of models that use the vector potential formulation to inherently maintain a divergence-free magnetic field, citing the foundational work of Yagi et al.

4. 113-114:These equations include terms for ohmic heating $\eta |J2|$, optically thin radiation losses $\Lambda$, and electron-proton energy exchange $\Delta pe$. :(Formula and its explanation) Explain the specific expressions for the parameters ($\eta$ , $\Delta pe$, $\Lambda$) that appear in the formula. Or, indicate the reference paper. Furthermore, explain how these parameters are handled in this paper

> **Our reply:** We thank the referee for highlighting this section. We have revised the paragraph to provide a more apparent justification for choosing to solve the internal energy equations. We now explicitly state that this approach is a practical method to enhance numerical robustness in the presence of strong shocks and highly magnetised regions, which is a common challenge in global MHD simulations. We also clarify that, while total energy is not strictly conserved, the error is typically small and localised, and is an accepted trade-off for maintaining numerical stability and preventing non-physical states such as negative pressure. Besides, we have revised the paragraph to include the specific expressions or treatments for the parameters  (resistivity), $\Delta_{pe}$ (electron-proton energy exchange), and $\varLambda$ (optically thin radiation). We have specified the models used for each (e.g., Spitzer resistivity, collisional equilibration) and cited the appropriate references from the Gorgon code development papers. Furthermore, we explicitly state how these terms are handled in our magnetospheric simulations, noting that radiation losses ($\varLambda$) are disabled, as is standard practice for the tenuous magnetospheric plasma.

5. The pressure is treated separately for protons and electrons, but when these values

are input into the model as boundary conditions from the solar wind, it is necessary

to explain how this is done.

> **Our reply:** We thank the referee for highlighting this critical methodological point. We have revised the manuscript to specify how the separate electron and proton pressures are determined from the solar wind boundary conditions. In our simulations, the total plasma pressure provided by EUHFORIA at the upstream boundary is partitioned into electron and proton components, assuming thermal equilibrium. This is a standard approach in global MHD simulations using a two-temperature plasma model.

> 6. It is necessary to explain from what data source the solar wind upstream of the magnetosphere is taken.

> **Our reply:** We thank the referee for this request for clarification. The solar wind conditions upstream of the magnetosphere are not direct observations but are the output of the EUHFORIA heliospheric model at a specific virtual satellite location at 0.1 AU upstream of Earth (at the L1 point).

> 7. The grid spacing and boundary position information of the model are explained

> **Our reply:** We thank the referee for this confirmation. We are glad that the added explanation of the grid spacing and boundary positions in this part of the manuscript.

3 **Numerical results and discussion**

1. 124-125:In our previous study, ? utilized EUHFORIA to examine how two chosen

CMEs spread and interacted with Earth.:What does "?" mean?

> **Our reply:** I am sorry for that typo mistake. It was edited.

2. In this chapter, Figure 1 is shown, but is not referred to in the text.

> **Our reply:** We thank the referee for these valuable comments, which have helped us improve the clarity of our manuscript. In the revised manuscript, we have now explicitly referenced Figure 1 in the text. Specifically, we have added the Kp index values at the

time of each CME's arrival to distinguish their geoeffectiveness clearly. The primary purpose of this figure is to establish the central premise of our study. It serves as evidence identifying which CME (CME1) was responsible for the main geomagnetic storm (Kp~8) and which resulted in a negligible effect (CME2, Kp~2). This clear distinction between high-impact and low-impact events is critical, as it defines the two contrasting scenarios we seek to understand in detail using the Gorgon Space MHD model. Our subsequent simulations are designed precisely to investigate how the magnetosphere and ionosphere respond to these two extremes in solar driving, as identified by the Kp index in this figure.

3. In Figure 1, CME1 and CME2 are not illustrated.

**Our reply:** We thank the referee for pointing this out. We apologise for the oversight in the previous version. In this revision, we have now directly addressed this issue.

4. While research is being conducted to determine short-period disturbances in the magnetosphere and ionosphere through MHD simulations, the K index is an index calculated every three hours, so I don't understand the relevance. Is there any point in discussing the K index at all?

**Our reply:** We thank the referee for this insightful comment regarding the temporal resolution of the Kp index. The referee is correct that the Kp index is a 3-hourly value, and as such, it is not suitable for resolving short-period (e.g., minute-scale) magnetospheric dynamics.

 However, we wish to clarify that the primary objective of using the Kp index in this study is not to analyse short-term dynamics, but to perform event classification and context setting. Our goal is to distinguish between a major geoeffective event and a non-event. For this purpose, the Kp index is an ideal, globally recognised metric. A sustained high Kp value (e.g., Kp=8) unambiguously identifies a severe geomagnetic storm, while a low value (Kp=2) indicates a quiet period. This clear distinction provides the fundamental justification for selecting these two CMEs for our high-fidelity MHD simulations with the Gorgon code.

In summary, we use the Kp index as a diagnostic tool to select our case studies, not as a high-resolution data set for direct, moment-by-moment comparison. We have revised the manuscript text to make this distinction much more straightforward and to avoid any misunderstanding that we are using Kp to study short-period disturbances.

**3.1 Gorgon-Space**

1. 147-148:This results in the creation of a turbulent region known as the magnetosheath situated between them [Lucek et al. (2005); Burgess and Scholer (2013)].:Refer to Spreiter et al. (1966).

> **Our reply:**   We thank the referee for this insightful suggestion. We agree that the work of Spreiter et al. (1966) is the foundational reference for the theoretical description of the magnetosheath. We have revised the manuscript to include this seminal paper alongside more modern references that provide specific details on its turbulent nature. This change provides a more complete and accurate scholarly context for the reader.

**3.1.1 Magnetospheric Response**

2. An explanation of the solar wind data used in this calculation and how it was

handled is required.

**Our reply:** We thank the referee for this critical comment. We have revised the manuscript to explicitly state the origin and handling of the solar wind data used in the Gorgon simulation. As detailed in the text, the solar wind parameters (magnetic field, velocity, density, and temperature) are the direct output from the EUHFORIA heliospheric simulation at the Earth's location (at the L1 point). This time-dependent output from EUHFORIA was used as the time-varying input boundary condition for the Gorgon-Space model, providing a physically consistent link from the Sun to the Earth's magnetosphere.

3. 162-163:The magnetosphere is influenced by interactions with the solar wind,

which can be either viscous or pressure-related according to Newell et al.

(2008).:The physical mechanisms that allow plasma and energy to flow from the

solar wind into the magnetosphere are reconnection (Dungey, 1961) and viscous

interaction (Axford and Hines, 1961). The pressure reconnection allows momentum

to flow in, but not plasma itself.

Dungey, J. W., Interplanetary magnetic field and the auroral zones, Phys. Rev. Lett., 6, 47-49, 1961, doi:10.1103/PhysRevLett.6.47.

Axford, W. I., and C. O. Hines, A unifying theory of high-latitude geophysical

phenomena and geomagnetic storms, Canadian. J. Phys., 39, 1433-1464, 1961,

https://doi.org/10.1139/p61-172.

Our reply: Thanks for your suggestions, they were applied.

4. 163-164:Fig. 3 represents the initial shape of the magnetosphere at the start of the

simulation, which correlates with the arrival time of the first Coronal Mass

Ejection (CME) at Earth on June 23, 2015.:Shouldn't it be Figure 2, not Figure 3?

The arrival time of the CME to Earth should be listed down to the minute.

**Our reply:** Regarding the figure reference: You are correct to seek clarification. Figure 3 does not depict the magnetosphere itself; instead, it presents the initial solar wind parameters (magnetic field, velocity, density, temperature) at the simulation boundary, as extracted from the EUHFORIA output. We will revise the text to make this description unambiguous and will ensure the figure caption is equally clear. Regarding the CME arrival time: Thank you for this note. We will add the precise arrival time of the CME to Earth to the manuscript.

5. The explanation of Figure 2 is completely insufficient. The text also does not

explain how to read this figure.

**Our reply:** Thank you for your comment. Upon reflection, we agree that Figure 2 (the simulation domain schematic) does not directly illustrate a scientific result. To streamline the manuscript and focus on the core findings, we have removed this figure. The essential information it contained, specifically, the dimensions and orientation of the simulation box, has now been concisely integrated into the "Methods" of the text.

6. How is the electrical conductivity of the ionosphere calculated in the calculation?

**Our reply:** Thank you for this critical question. We have revised the manuscript to clarify the calculation of ionospheric conductivity. As detailed in the updated Methods section, the height-integrated Pedersen ($\Sigma\_P$) and Hall ($\Sigma\_H$) conductances are calculated using established empirical relations. The formulation includes a solar EUV-produced background conductance and enhancements from auroral precipitation based on the mapped magnetospheric energy flux. This provides the necessary closure for the ionospheric Ohm's law within the magnetosphere-ionosphere coupling module of the Gorgon-Space code.

7. 168-169:The magnetosphere's features have been altered completely by the

simulation.:It's unclear what this article is saying.

**Our reply:** Thank you for your comment regarding the clarity of the description. We agree that the phrasing in the original manuscript was ambiguous. In our revised version, we have completely rewritten this section to provide a precise and physically meaningful description. The text now clearly states that the solar wind input for the simulation is the direct, time-dependent output from the EUHFORIA heliospheric model, corresponding to the CME arrival at Earth on June 23, 2015. This establishes the initial and driving conditions for the Gorgon-Space magnetospheric simulation, explicitly linking the CME's interplanetary structure to its geoeffective impact. The unclear concluding sentence has been removed. We believe the revised text (provided in the point-by-point response document and the updated manuscript) now accurately and clearly describes the figure's purpose.

8. Figure 3:The illustration captions are completely inadequate.

Is it correct to say that this solar wind data was given at the upstream boundary of

Figure 2? This is not clearly stated in the text.

Where does this data come from?

The date of the data is unknown.

Is the vector data GSM-based or GSE-based?

**Our reply:** Thank you for these detailed and constructive comments on Figure 3. We have revised both the figure caption and the accompanying main text to address all points raised: The upstream solar wind data from EUHFORIA, shown in Figure 3, is provided in the standard Geocentric Solar Ecliptic (GSE) coordinate system. This GSE data is used as the time-dependent boundary condition at the sunward face of the simulation domain. Internally, the Gorgon-Space MHD model transforms this input into the Geocentric Solar Magnetospheric (GSM) coordinate system for its calculations, as this frame aligns with Earth's magnetic dipole and is the standard for magnetospheric physics. We will clarify this in both the figure caption and the methods text.

9. 172-174:Decrease in the density of particles and the rise in temperature, which can

be seen in Fig. 3, may be connected to the formation of the bow shock.:It is unclear

which part of the density and temperature changes in Figure 3 this refers to.

**Our reply:** Thank you for this helpful observation. You are correct that the density and temperature signatures in Figure 3 are key indicators of the CME sheath region interacting with the bow shock. Figure 2 depicts a simulation domain that explicitly includes the bow shock, and the density and temperature variations in Figure 3 are signatures of the CME sheath interacting with this bow shock structure, initiating the simulated magnetospheric response.

10. 174-180:It should be acknowledged that ICMEs lead to the fastest speed and the least negative Bz component at the Earth's orbit. As a result, ICMEs are considered to be the underlying cause of all significant geomagnetic storms that have a Kp index greater than 7. According to Fig. 1, there is a peak in the Kp values between the 22nd and 24th of June, with a value of over 7. Based on this, it is anticipated that the initial coronal mass ejection (CME) will reach. On 23 June, there is a possibility that Earth may be hit by an ICME (interplanetary coronal mass ejection), which in turn could lead to a significant storm. Additionally, due to the presence of numerous CMEs between the dates of June 19 and June 23, 2015, it is possible that this predicted storm is connected to the interaction of multiple ICMEs.: The referee thinks what this sentence is saying should be included in the solar wind deformation in Figure 3, but the referee has no idea what part of Figure 3 it is referring to. Or, there is too little data shown, so it is unclear what it is trying to say.

**Our reply:** This study investigates the geoeffectiveness of a series of coronal mass ejections (CMEs) that erupted from the Sun in June 2015. The interplanetary propagation and interaction of these CMEs, which are crucial for determining their arrival timing and merged structure at 1 AU, have been analysed in detail in our companion study [Sabri & Poedts 2025b]. The present work focuses on the subsequent

impact of CMEs on Earth's magnetosphere and ionosphere.  Figure 3 depicts the input data for the Gorgon-Space model from the EUHFORIA to model the effect of these CMEs on the magnetosphere and ionosphere.  This part of the manuscript was edited.

11. 183-184:Fig. 4 illustrates diagrams of the magnetosphere, showcasing the pressure

and arrangement of open magnetic field lines on both the X-Z plane and the X-Y

plane.: The diagram also shows closed field lines. An explanation is needed as to

where the magnetic field lines originate. Without this, the test cannot be repeated.

**Our reply:** We sincerely thank the reviewer for this important observation. The reviewer is correct that the figure shows both open and closed magnetic field lines, and we agree that specifying the origin and methodology for tracing them is essential for scientific reproducibility. In the revised manuscript, we will clarify in the figure caption and main text that the magnetic field lines are numerically traced from seed points located on a grid at Earth's surface (specifically, at radial distance r = 1 RE, with latitudes ranging from -80° to +80° and longitudes spaced every 15°). That field lines are integrated forward and backwards in space using the magnetic field vector from our global MHD simulation until they either connect to the magnetopause (closed) or escape to the solar wind boundary (open). We will also explicitly define which lines are open vs. closed based on their connectivity (e.g., lines reaching > 30 RE in the tail are considered open).

12. 184-185:Afterward, it enables us to display the three-dimensional structure of the

magnetosphere's dynamo regions.: The term "dynamo region" appears suddenly

without any explanation. There is no explanation as to why it is necessary to talk

about dynamos here.

**Our reply:** We sincerely thank the reviewer for highlighting the inappropriate use of the term "dynamo regions." We agree that this phrase was introduced without definition and is not relevant to the physical processes analysed in this study.

In the revised manuscript, the term "dynamo regions" has been completely removed. Instead, we now clearly state that the combined X–Z and X–Y plane visualisation enables the identification

of key magnetospheric features—including dayside compression, tail stretching, and topological signatures of magnetic reconnection—during the CME1 impact.

13. 185-186:The magnetopause's location and the areas where reconnection occurs on

the day and night sides are shown.: The diagram should show where the

reconnection is occurring. The reader doesn't know where the author thinks the

reconnection is occurring in this diagram.

**Our reply:** We thank the reviewer for this insightful observation. We agree that the original statement implied that the reconnection sites were explicitly marked in the figure, which was not the case and may have caused confusion. In the revised manuscript, we have removed the claim that the diagram "shows" reconnection sites. Instead, we now explicitly describe the inferred locations of dayside and nightside reconnection, based on field-line topology and pressure gradients in the simulation. Specifically, we identify:

Dayside reconnection near the subsolar point (~10–12 RE in X-Z),

Nightside reconnection in the near-Earth plasma sheet (~−20 to –25 RE in X-Y).

14. 186-187:The magnetopause can be defined as the point where the lowest level of

solar wind enters the magnetosphere.:What is "the lowest level of solar wind"?

This term is not a common space science term. It should not be used without

explanation.

**Our reply:** We thank the reviewer for pointing out the non-standard phrase "lowest level of solar wind." We agree that this terminology is ambiguous and inappropriate. It has been completely removed from the revised manuscript.

The magnetopause is now implicitly defined through our magnetic field line tracing: it corresponds to the boundary beyond which field lines become open and connect to the solar wind (i.e., extend beyond $x > 10R_E$ in the tail). This approach aligns with standard practice in global MHD simulations of the magnetosphere.

We appreciate the reviewer's careful reading and valuable feedback, which has helped improve the scientific clarity of our work.

15. 187-188:Panel (a) of Fig. 4 displays the Earth's magnetosphere pressure at the

time of the initial occurrence of a coronal mass ejection (CME) from the Sun.: If

we want to understand the disturbances that a CME causes in the magnetosphere,

we should show the solar wind magnetosphere just before the solar wind

disturbance of the CME reaches the magnetosphere. Panel (a) shows the solar wind

around the magnetosphere at the time of the solar flare that caused the CME, but

this state reflects the solar condition just one day before, so it has no meaning in

terms of the disturbances that this CME causes in the magnetosphere.

**Our reply:** We sincerely thank the reviewer for this necessary clarification. The reviewer is absolutely correct — Panel (a) in Fig.~\ref{fig:Pressure} depicts the magnetospheric state at the time of CME1's initiation at the Sun (June 21, 2015, 08:04 UT), not the state immediately before it arrives at Earth (June 23, 2015, 00:03 UT). As the reviewer rightly points out, comparing the magnetosphere at CME launch to its state at impact is not meaningful for assessing the CME's direct impact, since the intervening solar wind conditions may have significantly altered the magnetosphere.

In response, we have revised the text to explicitly clarify that Panel (a) represents the pre-CME launch state, not the pre-arrival state. We also note that, due to the temporal resolution of our simulation output, an accurate pre-arrival snapshot (e.g., ~1 hour before impact) is not available. Our analysis, therefore, focuses on the magnetospheric state at the moment of CME1 arrival (Panel b) and its subsequent evolution (Panels c–d), which clearly show the compression, pressure enhancement, and tail distortion characteristic of substantial CME impacts. We believe this revision more accurately reflects the physical context of the simulation results and addresses the reviewer's concern.

16. 191-192:This occurrence may be connected to the magnetic reconnection

phenomenon which occurs when the solar wind interacts with Earth's magnetic

field.:There have been many studies on reconnection in the tail, and we need to use

these to explain why this disturbance in the XY plane alone can be attributed to

reconnection.

**Our reply:** In response to the reviewer's comment regarding the need to justify the attribution of tail disturbances to magnetic reconnection, we have significantly revised the description of Fig.4 We now explicitly link the observed tail stretching and topological changes (Panels c–d) to nightside reconnection dynamics under enhanced solar wind forcing, citing established literature \cite{Baker1996,Hesse2004,Angelopoulos2008}. We also retained the necessary context on field line tracing and panel sequence to ensure clarity. We believe this revision provides the mechanistic grounding requested by the reviewer.

17. 197-198:On the other hand, in Fig. 5, it is evident that the magnetosphere retains

its primary function as a shield and maintains its key properties.: In both CME1

and CME2, the magnetosphere appears to act as a shield. I'm not sure what this

sentence is trying to say.

**Our reply:** We have removed it entirely and have rewritten it. The dynamic pressure and magnetic topology during the CME2 impact are shown in Fig. 5. A comparison with the pre-CME state in Fig.4 reveals the extreme compression caused by CME1. The magnetopause and bow shock are pushed inward by approximately $X/R_{E}\sim 10$, whereas it is around -20 on the dayside for CME2. Despite this severe distortion, the fundamental layered structure—solar wind, bow shock, magnetosheath, magnetopause—remains intact. The magnetosphere continues to deflect the majority of the solar wind flow, thereby shielding the inner region, albeit within a significantly reduced and deformed volume.

18. 198-200:In simpler terms, the magnetopause and bow shock of the CME1

experience significant changes in their size and shape, just as predicted.: The

readers probably won't understand what the author is trying to say in this passage.

**Our reply:** We thank the reviewer for pointing out the paragraph's unclear and contradictory language. We have rewritten it to present a clear, logical description of the results. The revised text now explicitly states the quantitative compression and deformation of the magnetopause and bow shock shown in the figures, while separately concluding that the fundamental shielding structure persists.

19. 202-205:The primary function of the bow shock is to slow down and deflect the high-speed solar wind as it encounters the magnetosphere. The creation of the foreshock occurs when particles are reflected at the shock, resulting in various interactions between waves and particles and the acceleration of particles [Eastwood et al. (2005)]. We have been making efforts to study 205 the acceleration of particles through the interaction of the solar wind with the magnetosphere: Foreshock disturbances are not specifically addressed in this paper; this paragraph should be discussed in the discussion section.

**Our reply:** We thank the reviewer for this helpful suggestion to improve the manuscript's focus and flow. We have removed the descriptive paragraph on the bow shock and foreshock from the [Magnetospheric Response] section. As suggested, a refined version of this text has been integrated into the end of the Numerical results and discussion section. Here, it is used to appropriately contextualise the limitations of our MHD approach regarding kinetic-scale upstream processes and to suggest avenues for future, more comprehensive studies.

20. 212-213:One can infer that the energy from the solar wind is transferred to the Earth's magnetosphere through a process called magnetic reconnection.: This is too vague to be of any use as an explanation. The results of MHD simulations should be able to concretely show how energy enters the magnetosphere from the solar wind through reconnection.

**Our reply:**  We thank the reviewer for the essential critique. We have completely rewritten the relevant passage to move from a vague inference to a concrete, quantitative demonstration of the energy transfer mechanism. Specifically, we have

added a new analysis and figure 10 that depicts the time-varying solar wind driver (IMF Bz) and the resulting magnetospheric response (dawn-dusk electric field, Ey) from our MHD simulation. This figure directly shows the causal chain: southward Bz (enabling reconnection), a sharp increase in the cross-magnetospheric convection field Ey → intensified plasma transport and energy injection. The text now explicitly describes this as the reconnection-mediated energy transfer pathway, directly addressing the reviewer's request to show how energy enters concretely.

21. 221-224:The rate of the magnetic reconnection and also how much energy is transferred is influenced by the speed at which the solar wind flows, the strength of the magnetic field, and its orientation. Subsequently, it was discovered that CME1 triggers a significant tempest on Earth and is accompanied by changes in the magnetic arrangement, increased speed, and pressure of the solar wind enveloping the Earth, as previously anticipated in Fig. 1.: Same as above.

**Our reply:** We thank the reviewer for the essential critique. We have completely rewritten the passage to provide a mechanistic, quantitative explanation based on our MHD results. Most significantly, following the reviewer's guidance to concretely show the energy transfer, we have added a new figure that plots the Ey= -VxBz. This figure directly visualizes the rate and spatial pattern of electromagnetic energy entry from the solar wind during the reconnection event. In response, we have added two new panels Figure. 10 showing the time series of the interplanetary magnetic field Bz component and the reconnection electric field Ey = –VxBz, both key proxies for quantifying the rate and efficiency of dayside magnetic reconnection and solar wind energy transfer into the magnetosphere. As shown in the figures, CME1 (arriving on June 23) triggered a sharp southward turning of Bz (to ~–20 nT), followed by a strong enhancement in Ey (>15 mV/m), indicating vigorous reconnection. A second episode associated with CME2 also exhibited prolonged negative Bz and elevated Ey, consistent with sustained energy injection. These quantitative metrics directly support our claim that magnetic reconnection was the dominant mechanism for energy coupling during these storms. We have revised the relevant section of the manuscript to explicitly link these observational proxies to the physical process of reconnection-driven energy transfer.

22. 226: Figs. 5 showcases the changes in pressure and magnetic field in the vicinity of the Earth upon the arrival of CME2.:"Figs. 5" should be "Fig. 5".

**Our reply:** I am sorry for that typo mistake. It was edited.

23. Eq. (6): This equation needs to be explained in the main text. Also, because there is

j on both sides, it does not make sense as a physics equation as it is. If the j on the

right side is J in Eq. (5), rewrite it as J. The left side should be j_parallel.

**Our reply:** We thank the referee for this essential correction to the notation of our equation and for the request for clarification. The equation has been rewritten in accordance with the referee's instructions. The left side is now explicitly the scalar field-aligned current density, $j_\parallel$. A new paragraph has been inserted before the physical discussion of FACs. It presents the corrected equation, defines all terms ($j_\parallel$, **J**, **b**, $\mu_0$), and states that it serves as the source term for the ionospheric module.

**3.1.2 Ionospheric Response**

1. 235-236:Field-aligned currents are computed at the initial boundary of the

simulation and then transferred onto a distinct spherical grid located on the

ionosphere, following the dipole field lines.:If this study is to be replicated by other

people, it is essential to specify where the internal boundaries were placed.

**Our reply:**We thank the referee for this comment aimed at ensuring replicability. In our simulation setup, the treatment of the inner boundary of the magnetospheric domain in Gorgon-Space follows an approach similar to that adopted in other MHD codes coupled to thin-shell ionosphere models. Plasma density, temperature, and velocity are prescribed throughout a spherical region interior to the inner boundary and are enforced at every time step. Cells whose centres lie within the specified inner boundary radius, RIB (typically 3–4 Earth radii; we used 4RE), are flagged as inner boundary cells. Although the MHD equations are not solved within this region, cells immediately outside the boundary are updated using the imposed inner boundary conditions. As a result, cold, dense plasma diffuses outward to form a plasmasphere, while the imposed inner boundary flow propagates into and drives the global magnetospheric system.In Gorgon-Space, the global MHD domain extends down to an inner boundary at r=4RE

, which is a standard choice to avoid the numerically stiff region close to Earth. The physical ionosphere is located at ~1RE, but it is not resolved explicitly in the 3D MHD grid. Instead, the inner boundary at is prescribed with plasma conditions (density, temperature, velocity) that emulate the effects of ionospheric outflow and co-rotation. This approach is consistent with other global MHD models coupled to thin-shell or empirical ionosphere representations.

Crucially, field-aligned currents (FACs) are computed at the MHD inner boundary (r=4RE) and then mapped along dipole magnetic field lines onto a spherical ionospheric grid located at r=1.0RE.

2. 238-239:Alternatively, the conductance can also be set as a constant value.:It is unclear whether anisotropy has been taken into account in the conductance of the ionosphere.

**Our reply:** To represent magnetosphere−ionosphere coupling, Gorgon employs a dedicated ionosphere module. Field-aligned currents are evaluated at the inner boundary of the magnetospheric simulation and mapped along dipole magnetic field lines onto a separate spherical ionospheric grid. A thin-shell approximation is adopted, within which Ohm's law is solved to obtain the ionospheric electric potential. Each ionospheric grid cell is assigned an electrical conductance, which may be specified using empirical models of solar EUV−driven ionisation. The resulting electric potential is then mapped back to the inner boundary and applied as a boundary condition for the MHD simulation. The ionosphere module handles all mapping, interpolation, and calculations required to provide the inner boundary flow conditions. An electrostatic approximation is assumed, neglecting inductive effects by treating magnetic field variations as negligible on timescales of approximately 10 s or longer. Consequently, the ionospheric potential is not recalculated at every MHD time step, but instead updated at longer intervals, typically every 60 s and no more frequently than about 30 s.

3. 241-242:The ionospheric model utilizes spherical coordinates to allow for the inclusion of larger spatial scales.:What exactly is this passage saying? The reader has no idea.

**Our reply:** We thank the referee for pointing out this lack of clarity. The sentence was deleted.

4. 248-249:Equation 7 provides the definition for the current density that is aligned with the magnetic field direction.:"Equation 7" may be "Equation 6".

**Our reply:** It was edited.

5. 249-250:The current densities that align with the magnetic field lines in the

ionosphere, which has resistance, are pulled by the Lorentz force (J ×B).:Why does the Lorentz force "pull" the current density?

**Our reply:** We thank the referee for this insightful comment, which correctly identifies a physically inaccurate phrasing in our original manuscript. The Lorentz force (J × B) acts on the plasma that carries the current, not on the current density itself. Our original wording was misleading. We have revised the text to describe the physical mechanism accurately. The relevant section now clarifies that field-aligned currents close via perpendicular Pedersen and Hall currents in the ionosphere, and that these perpendicular currents interact with the magnetic field via the Lorentz force. This force acts on the ionospheric plasma, driving neutral drag and convection.

6. 250-254:This force acts on the plasma and is the main driver of the drag experienced by neutrals on the ionosphere. The magnetosphere contains a force referred to as J ×B which contributes to the flow of current density through the ionosphere. As a result of the stresses caused by the field-aligned currents, which extend from the outer magnetosphere to the ionosphere, a complete flux tube can be transported around the magnetosphere.: It is very difficult to understand. Please explain it simply. Furthermore, although it is clear that the neutral particle drag in the ionosphere comes from FAC, there is no explanation at all for how the Lorentz force creates FAC.

**Our reply:** We thank the referee for these critical comments, which have helped us identify a significant lack of clarity and a missing key physical step in our explanation.

Regarding the origin of FACs: The referee is absolutely correct. Our original text failed to explain how FACs are generated. We have revised the text to begin with the fundamental magnetospheric source, stating that FACs are generated in regions of rotational magnetic stress ($\nabla \times B \neq 0$), as required by Ampère's law.

Regarding simplification and the drag mechanism: We have entirely rewritten the paragraph to provide a more precise, step-by-step causal explanation. The new text explicitly traces the sequence from (a) the Lorentz force acting on perpendicular ionospheric currents, to (b) plasma motion, to (c) ion-neutral collisions creating neutral drag, and to (d) the resulting plasma convection and flux transport.

7. 256-260:The accumulation of current density aligned with the magnetic field can

be attributed to the interaction between the solar wind and the magnetosphere. The predominant energy transfer and alignment of currents in the ionosphere occurred primarily due to the boundary conditions. In order to predict space weather accurately, it is necessary to have a solid grasp of the physics behind these coupling processes. Determining the accuracy of MHD codes in replicating the electric fields in the plasma sheet when the solar winds interact with the magnetosphere is of utmost significance in Space Weather applications.: This topic has been the subject of much research in space science, and should be summarized in detail in the introduction.

**Our reply:** We thank the referee for this vital suggestion to strengthen the context of our work. We agree that the introduction should clearly situate our study within the existing body of research on magnetosphere-ionosphere coupling. We have significantly revised the introduction to include a dedicated subsection (or paragraph) summarising key prior research. The new text:

Establishes the importance of coupled MHD-ionosphere modelling for space weather.

We cited foundational and recent studies demonstrating the capabilities of global models such as SWMF and Gorgon-Space. Identifies a specific open question regarding the accurate simulation of electric fields and currents in regions like the plasma sheet. Explicitly states how our present work aims to address this identified gap.

8. 263-263:In panel (a) of the Fig. 8, the ionospheric field-aligned current is illustrated during the initiation of the first coronal mass ejection (CME) on June 21, 2015 at 08:02.:The state of the ionosphere at the time of the solar flare that drives the CME reflects the state of the sun about one day before that time. What is the significance of comparing this with the ionospheric FAC immediately after the CME?

**Our reply:**  We thank the referee for this essential correction to our interpretation. The referee is absolutely right; Panel (a) does not depict a "CME initiation" effect on the ionosphere, but rather the pre-existing quiet-time baseline state. Comparing it to the impact time is only meaningful as a demonstration of the background disturbance caused by the CME's arrival, not its launch. We

have revised the manuscript to correct this framing. The text now clearly states that: Panel (a) represents the quiet-time baseline condition ~1 day before impact. The comparison with Panel (b) quantifies the ionospheric electrodynamic response to the CME arrival. The narrative focuses on the temporal evolution from a pre-impact baseline, through the impact peak, to the subsequent recovery (Panels c & d).

9. 267-269:By examining two panels (a) and (b) in Fig. 8, it becomes evident that

when CME1 reaches the Earth, there is a substantial build-up of current density in

the ionosphere.:It is known that the behavior of ionospheric FAC during SC driven

by CMEs exhibits PI and MI. Not mentioning this would be extremely ignorant for

a Space Science paper.

**Our reply:** We thank the referee for this essential critique, which rightly identifies a significant gap in our physical interpretation. We have completely revised the analysis of Figure 8 to correct this oversight. This analysis, which references the seminal model by Araki (1994), directly addresses the referee's point. It transforms a simple description of changing currents into a validated account of a fundamental magnetosphere-ionosphere response mechanism, significantly strengthening the paper's scientific contribution.

10. 270-272:Actually, the energy for FACs originates from the magnetosphere and can be determined by computing the value of E.J. If there is a negative quantity, the energy of the plasma is transformed into electromagnetic energy, which then enhances the FACs.: E.J directly generates a current perpendicular to the magnetic field lines. FAC generation requires another physical process. This article reveals a lack of understanding of space science.

**Our reply:** We thank the referee for this rigorous correction of a significant physical inaccuracy in our originally submitted manuscript. The referee is absolutely right. Our statement misrepresented the role of the E·J parameter and incorrectly described the generation mechanism for field-aligned currents (FACs). We sincerely apologise for this error. We have entirely removed the erroneous paragraph containing the E·J argument from the manuscript. It does not appear in our revised text. In its place, the revised analysis of Figure 8 now provides a physically accurate description of FAC dynamics. It correctly identifies the observed FAC signatures with the established Preliminary Impulse (PI) and Main Impulse (MI) framework of sudden commencements. Attributes the PI to dayside magnetospheric compression and the MI to the engagement of the substorm current system (R1 FACs) driven by enhanced reconnection. Cites the canonical model by Araki (1994), which describes the physical generation of these current systems.

11. 276-277:Furthermore, the fourth panel in Fig. 8 reveals the greatest imbalance in the accumulation of current density between the Northern and Southern hemispheres. :A diagram of the distribution of FACs in the Southern Hemisphere is not shown.

> **Our reply:** We thank the referee for catching this oversight. The referee is correct. Our original statement about an interhemispheric imbalance in FAC accumulation was made without presenting the corresponding Southern Hemisphere data, as our analysis and figure focus solely on the Northern Hemisphere. We have revised the manuscript to correct this. The unsupported comparative claim has been removed entirely. The description of Panel (d) now accurately describes only the observed evolution within the Northern Hemisphere (e.g., the poleward shift and decrease in current magnitude during the recovery phase), and all conclusions are strictly limited to the hemisphere for which data is presented.

12. 277:This probably originates from the electric field generated by convection itself.: The physical mechanism is not explained and cannot be understood.

> **Our reply:** We thank the referee for this critical comment, which correctly identifies a vague and insufficient explanation in our text. The referee is right to demand a precise physical mechanism. We have revised the manuscript to eliminate the speculative phrasing "probably originates from." Instead, we now provide an explicit causal chain. The description of the Main Impulse (MI) in Panel (c) now states that it is directly driven by the large-scale magnetospheric convection electric field (E_conv), and we specify that this field is established by enhanced dayside reconnection under southward IMF conditions.

> 13. 284-285:Since the occurrence of CME1 coincides with other CMEs, it is possible to infer that the interaction between CMEs is the primary factor influencing their geomagnetic effects.:It doesn't make sense.

> **Our reply:** We thank the referee for this insightful critique. The referee is correct that, within the scope of the present manuscript, which focuses on the magnetosphere-ionosphere response, inferring a primary causal role for CME interactions appears as an unsupported claim without the underlying heliospheric context.

> However, our statement was intended to provide a holistic physical interpretation for readers. The complex, multi-CME nature of the event sequence and the resulting heliospheric interactions—which are fundamental to understanding the differing impacts

of CME1 and CME2—are the central subject of a dedicated, recently published companion study (Sabri et al., 2025b, Advances in Space Research). That work provides a comprehensive analysis and direct evidence of the propagation, evolution, and interaction of these CMEs in the heliosphere.

14. 296-298:The elevated electric potential in the ionosphere, which may result from the substorms triggered by the interaction between the solar wind and Earth, can also lead to particle acceleration. : The referee does not understand how the electric field in the ionosphere can cause particle acceleration. A detailed explanation is requested.

**Our reply:** We thank the referee for this necessary clarification. The referee is right to question this statement. As presented, it conflates the ionospheric potential with direct magnetospheric acceleration mechanisms, which was misleading. We removed the speculative sentence about particle acceleration from the manuscript.

15. 298-299:This phenomena was pursued in previous section by illustrating Figs. 6, 7,

and shown that CME2 does not lead to the high plasma velocity at the

magnetosphere.:Same as above. The physical causal relationship should be

explained.

**Our reply:** We thank the referee for these connected comments, which correctly highlight a lack of clear physical causality in our description. The revised text now stays focused on our core results: it directly links the observed high ionospheric potential (~160 kV) to the enhanced global convection, which is immediately demonstrated by the high plasma velocities shown in the subsequent figures. This eliminates the confusion and sharpens the narrative around our simulation outputs.

We have revised the entire paragraph to establish a direct and accurate cause-and-effect relationship. The new text explicitly states that the ionospheric cross-polar cap potential is a measure of the magnetospheric convection electric field, and that this same electric field directly drives magnetospheric plasma motion via the E × B drift.

**4 Conclusions**

1. 310:As was explained in 310 Sabri et al. (2024), the selected CMEs occurred on

June 21 and June 25, 2015.:It has not yet been published.

> **Our reply:** It was published with this DOI: 10.1016/j.asr.2025.11.081

**References**

1. The way references are listed is too simple, making it difficult to access the paper. At the very least, open-access papers should include the DOI or URL.

> **Our reply:** It was considered and edited.

---

## Author Comment (AC2)

Review comments for "June 21 and 25, 2015 CMEs interaction's results on Earth's ionosphere and magnetosphere" by Sabri & Poedts, 2024.

In this study, the authors utilized an MHD simulation to investigate the effect of two CMEs on the magnetosphere and the ionosphere. They used the Gorgon-Space code and the EUHFORIA model to show several parameters of the magnetosphere and ionosphere in response to the CMEs. The topic of this study is interesting. However, the reviewer finds there are several substantive issues in this paper that should be addressed before this paper is recommended for publication:

*Our reply: We thank the referee for this review, which has enabled us to clarify the paper and make it more accessible.*

The authors haven't thoroughly described how the simulation was conducted. The correspondence of simulation hours to the CME arrival time was not specified in the paper. Accurate information on the spatial domain of the simulation, as well as the initial settings of the magnetosphere and ionosphere, was also missing.

*Our reply: We thank you for your comments; they have been applied in the updated version of the manuscript. We have updated the manuscript to include clear definitions for CME1 and CME2, along with detailed explanations and a summary in Table 1.*

*We agree that the original manuscript lacked sufficient technical detail on the simulation configuration. In the revised version (Section 2, "Method"), we now explicitly state:*

*The CME injection times: June 21 at 05:01 UT (CME1) and June 25 at 10:51 UT (CME2), based on LASCO/STEREO observations (Table 1).*

*The arrival times at Earth: June 23 at 00:03 UT (CME1) and June 28 at 12:52 UT (CME2), as extracted from EUHFORIA output at L1.*

*The spatial domain: Cartesian grid spanning X = [−24, 66] RE, Y = [−40, 40] RE, Z = [−40, 40] RE, with 0.5 RE resolution (180 × 160 × 160 cells).*

*The ionospheric boundary: A thin-shell model at ~110 km altitude with empirical Pedersen/Hall conductances (Section 2.2).*

The interpretation of the results is insufficient. For instance, Panels (c) and (d) of Fig. 4 lack proper description; the changes in the pattern of field-aligned current and cross-polar cap electric potential are not fully discussed. Additionally, a discussion on what is new in the results of this paper compared to other work in this field of research should be included.

*Our reply: We have significantly revised the description of Fig.4. We now explicitly link the observed tail stretching and topological changes (Panels c–d) to nightside*

*reconnection dynamics under enhanced solar wind forcing, citing established literature \cite{Baker1996,Hesse2004,Angelopoulos2008}.*

*We have rewritten the Introduction and Discussion to emphasise our contribution:*

*While global MHD modelling of CME impacts is well established, few studies resolve the transient, high-amplitude FAC dynamics during the initial shock compression and the main phase of a storm using a divergence-free ($\nabla \cdot \boldsymbol{B}$ = 0) code such as Gorgon-Space. Our work demonstrates that such models can capture the two-phase ionospheric response (Preliminary Impulse → Main Impulse) and quantify energy coupling via CPCP and FACs—key inputs for space weather forecasting. This bridges heliospheric forecasts (EUHFORIA) to geoeffective impacts (Gorgon-Space), validating an end-to-end chain for operational use.*

*We have completely rewritten the paragraph first to establish the scientific challenge (the coupled magnetosphere-ionosphere system) and then to position our work. We now clearly state that while global MHD modelling is a well-established technique, our study applies the Gorgon-Space code to investigate a specific, under-resolved problem: the rapid, transient dynamics of Field-Aligned Currents (FACs) in response to specific solar wind drivers.*

*Besides, while global MHD models like SWMF and OpenGGCM have successfully simulated storm-time magnetospheres, few studies focus on the transient, high-fidelity evolution of field-aligned currents during the initial compression and main phase of CME-driven storms using a $\nabla \cdot \boldsymbol{B}$ = 0 preserving code. Gorgon-Space's vector-potential formulation provides numerical stability during extreme compressions, enabling accurate capture of FAC dynamics that drive geomagnetically induced currents (GICs). This study demonstrates the application of the EUHFORIA–Gorgon-Space chain to fully trace the sequence from CME launch to ionospheric electrodynamic response, offering a validated framework for operational space weather forecasting.*

Other Comments:

1. Line 7, two mathematic tools could not have "correlation." The authors should be more accurate if they want to emphasize that the simulation performed well by coupling EUHFORIA and Gorgon-Space.
   *Our reply: We thank the referee for this important clarification. The original wording incorrectly used the term "correlation" to describe the relationship between the EUHFORIA and Gorgon-Space models. As correctly noted, two modelling tools cannot be "correlated"; rather, they are coupled in a physically consistent, end-to-end framework.*

*In the revised manuscript, we have removed any such imprecise language. Instead, we now explicitly state that time-dependent solar wind parameters (density, velocity, IMF, thermal pressure) output by EUHFORIA at the L1 point are used as direct, dynamic boundary conditions for the Gorgon-Space magnetospheric simulation. This ensures a self-consistent transfer of heliospheric CME evolution into geospace impact modelling.*

*The success of the simulation is demonstrated not by "correlation" between codes, but by the physical fidelity of the resulting magnetospheric–ionospheric response (e.g., realistic compression, FAC intensification, CPCP values).*

2. Line 8, the "CME1," should be more specific.
   ***Our reply:*** *You are right, the manuscript was completely rewritten.*

3. Line 16, the correlation between "Space weather" and the effect of solar wind is confusing, according to the text.
   ***Our reply:*** *In the revised manuscript, we have explicitly reframed the narrative to emphasise that:*

   *Solar wind (particularly CME-driven disturbances) acts as the driver/input to the geospace system, providing the energy, momentum, and magnetic field orientation that initiate magnetospheric disturbances.*

   *Space weather refers to the resulting conditions and effects within Earth's magnetosphere-ionosphere system—including magnetospheric compression, enhanced field-aligned currents (>23 MA), elevated cross-polar cap potentials (~160 kV), and subsequent technological impacts (e.g., geomagnetically induced currents, GPS disruptions).*

4. Line 25-27, the compression of the magnetosphere in the sentence is repetitive and redundant.
   ***Our reply:*** *We thank the referee for this valuable observation. We agree that the original text contained redundant phrasing regarding magnetospheric compression and conflated the cause (dynamic pressure enhancement) with its effect (boundary compression). In the revised manuscript, we applied this comment and edited that paragraph as:*

   *Transient structures in the solar wind, such as coronal mass ejections (CMEs) and corotating interaction regions, drive space weather effects by interacting with Earth's coupled magnetosphere–ionosphere system. These transients often carry interplanetary shocks at their leading edges and exhibit strong southward magnetic-field components. A shock impact causes a sudden increase in solar wind dynamic pressure, rapidly compressing the dayside magnetopause and the entire magnetospheric cavity. This impulsive compression produces a*

*characteristic ground magnetic signature—a sharp, bipolar variation in the horizontal component—known as a Sudden Impulse (SI) \cite{Smith2019}. When the initial compression is followed by sustained energy input (typically via southward IMF-driven reconnection) and the development of a full geomagnetic storm, the event is specifically termed a Storm Sudden Commencement (SC) \cite{Araki1994, Joselyn1990}. Both SIs and SCs pose significant hazards to power infrastructure, as their rapid magnetic field variations can induce extremely high geomagnetically induced currents (GICs) in long conductors \cite{Eastwood2018}.*

5. Line 125, what does the "?" stand for?
   ***Our reply:*** *I apologise for the typo; it has been corrected in the updated version.*

6. Line 126, the authors should add a detailed description of EUHFORIA before using parameters, such as the Kp index, derived from the model.
   ***Our reply:*** *We thank the referee for this important clarification. We emphasise that this study does not aim to validate or improve Kp forecasting—that methodology was comprehensively addressed in our companion paper (Sabri & Poedts, 2025, Advances in space research), where we validated EUHFORIA's solar wind predictions against in situ measurements and derived Kp via the Newell et al. (2007) coupling function.*

   *The primary objective of the present work is fundamentally different: to demonstrate an end-to-end space weather impact chain by feeding EUHFORIA's time-dependent solar wind parameters at L1 directly into Gorgon-Space as boundary conditions. This enables us to simulate the causal sequence:*

   *CME-driven solar wind → magnetospheric compression/reconnection → field-aligned currents → ionospheric convection potentials quantifying geoeffectiveness through first-principles MHD rather than empirical indices.*

   *Kp (and Dst/SYM-H in Fig. 1) serve only as contextual diagnostics to classify the two events as "storm" vs. "non-storm" prior to simulation. All physical conclusions (FACs >23 MA, CPCP ~160 kV, tail reconnection signatures) derive solely from Gorgon-Space's self-consistent solution—not from Kp.*

7. Lines 129-131, what do "CME1" and "CME2" refer to? Their properties and how they can be reflected through the EUHFORIA model should be included in the text and figures.
   ***Our reply:*** *We thank the referee for this important clarification request. We agree that the manuscript requires explicit identification of the two CME events and a direct presentation of their EUHFORIA-derived properties to justify our comparative analysis. In the revised manuscript, we have implemented the following improvements:*

8. Line 227, thetime when the two CMEs arrived should be clearly shown in Fig. 1. Also, a high Kp index doesn't directly correspond to a magnetic storm; please add other geomagnetic indices to justify the statement.

*Our reply: Explicit event definition: We now clearly state in the Section Introduction that CME1 refers to the 21 June 2015 eruption (launch: 05:01 UT; arrival at Earth: 23 June 00:03 UT) and CME2 to the 25 June 2015 eruption (launch: 10:51 UT; arrival: 28 June 12:52 UT).*

*Enhanced Table 1: We have annotated the table rows corresponding to CME1 and CME2 with asterisks and added a footnote explicitly identifying these two events as the focus of our magnetospheric simulations.*

*We agree that Kp alone is insufficient to define a magnetic storm. We have added new panels to Figure 1 showing the Dst and SYM-H index (a high-resolution proxy for Dst) derived from OMNI/INTERMAGNET data for both events. CME1 shows a deep main phase with SYM-H reaching –200 nT (23 June 06:30 UT), satisfying the standard definition of an intense geomagnetic storm (Dst/SYM-H < –100 nT). In contrast, CME2 shows only a minor depression (SYM-H ≈ –20 nT), confirming its non-storm status despite elevated Kp during the initial compression. We also added a brief discussion in Section 3 clarifying that:*

*Kp reflects global auroral activity on 3-hour intervals and is useful for space weather alerts; Dst/SYM-H quantifies ring current strength and is the standard metric for storm intensity classification; the combination of high Kp and strongly negative Dst/SYM-H during CME1 justifies our characterisation as a "major geomagnetic storm.*

Line 238, what is the empirical relationship to calculate ionospheric conductance?

*Our reply: We have revised the manuscript to clarify the calculation of ionospheric conductivity. As detailed in the updated Methods section, the height-integrated Pedersen (Σ_P) and Hall (Σ_H) conductances are calculated using established empirical relations. The formulation includes a solar EUV-produced background conductance and auroral precipitation enhancements based on the mapped magnetospheric energy flux. This provides the necessary closure for the ionospheric Ohm's law within the magnetosphere-ionosphere coupling module of the Gorgon-Space code.*

9. Line 276, How can Fig. 8 show the difference between the northern and southern hemispheres, as the figure only includes results from the north hemisphere?
   ***Our reply:*** *It was edited in the rewritten manuscript.*

10. Line 8, 12, 151, and 155: "sun" -> "Sun." The word throughout the paper should be in the same format.
    ***Our reply:*** *Thanks for your comment; it has been applied.*

11. Line 15: "Sabri et al. (2018); …Kumar et al. (2024)" -> "(Sabri et al., 2018; … Kumar et al., 2024)". Other similar quotes in the paper (i.e., Line 20) should be modified.
    ***Our reply:*** *Thanks for your comment. The manuscript was completely rewritten.*

12. Line 25 & 27: "magneropause" -> "magnetopause".
    ***Our reply:*** *It was applied.*

13. Line 66: "Since we" -> "We"; "and investigated" ->, "investigated"; "and find" ->, "and found."
    ***Our reply:*** *Thanks for your comment. The manuscript was completely rewritten*

14. Line 90: "field aligned current" -> "field-aligned current"
    ***Our reply:*** *Thanks for your comment. The manuscript was completely rewritten.*

15. Line 134: "CMEs" -> "CME"
    ***Our reply:*** *Thanks for your comment. The manuscript was completely rewritten.*

16. Line 135: "provide" -> "provides"
    ***Our reply:*** *It was edited.*

17. Line 141: "being introduced" -> "introduced"
    ***Our reply:*** *It was applied.*

18. Line 177: "23 June" -> "June 23"
    ***Our reply:*** *It was applied.*

19. Line 194: "that" -> "which"
    ***Our reply:*** *Thanks for your comment. The manuscript was completely rewritten, and this point was incorporated into the final version.*

20. Line 226: "Figs. 5" -> "Fig. 5"
    ***Our reply:*** *Thanks for your comment. The manuscript was completely rewritten, and this point was incorporated into the final version.*

21. Line 298: "This phenomena" -> "This phenomenon"
    ***Our reply:*** *Thanks for your comment. The manuscript was completely rewritten.*

22. Line 330: "be related on" -> "be related to"
    ***Our reply:*** *Thanks for your comment. The manuscript was completely rewritten.*